# Atomically precise ultrasmall copper cluster for room-temperature highly regioselective dehydrogenative coupling

Teng Jia[1], Yi-Xin Li[1], Xiao-Hong Ma[1], Miao-Miao Zhang[1], Xi-Yan Dong [1,2], Jie Ai[1] & Shuang-Quan Zang [1] ✉

Three-component dehydrogenative coupling reactions represent important and practical methodologies for forging new C−N bonds and C−C bonds. Achieving highly all-in-one dehydrogenative coupling functionalization by a single catalytic system remains a great challenge. Herein, we develop a rigid-flexible-coupled copper cluster $[Cu_3(NHC)_3(PF_6)_3]$ ($Cu_3NC^{(NHC)}$) using a tridentate N-heterocyclic carbene ligand. The shell ligand endows $Cu_3NC^{(NHC)}$ with dual attributes, including rigidity and flexibility, to improve activity and stability. The $Cu_3NC^{(NHC)}$ is applied to catalyze both highly all-in-one dehydrogenative coupling transformations. Mechanistic studies and density functional theory illustrate that the improved regioselectivity is derived from the low energy of ion pair with copper acetylide and *endo*-iminium ions and the low transition state, which originates from the unique physicochemical properties of the $Cu_3NC^{(NHC)}$ catalyst. This work highlights the importance of N-heterocyclic carbene in the modification of copper clusters, providing a new design rule to protect cluster catalytic centers and enhance catalysis.

Propargylamines including C1-propargylamines are essential building blocks for organic synthesis, the production of pharmaceuticals and compounds of substantial synthetic value industry[1–3], including oxazolidines[4], cavidine[5], 6-amino-pyridones[6], solifenacin[7], apomorphine[8], nascapine[9] and so on. Thus, much investigation has been devoted to the synthesis of propargylamine and C1-propargylamine compounds. For synthesizing propargylamines (Fig. 1a, b), researchers commonly exploit the different ways to form metal acetylides, such as the terminal acetylenes activation of bases or transition-metal salts, then the addition of the metal acetylides to imines to synthesize propargylamines[10–19]. For synthesizing C1-propargylamines (Fig. 1c), the copper compounds as representative catalysts could catalyze the redox-A³ coupling reaction under specific conditions[3]. The α-amino acid[20–22], simple amine[23–25] and tertiary amine[26,27] as representative amine reactants were used in the organic transformations under the hard conditions by Li[20], Seidel[21,23], and Ma[25]

et al. Due to the unique physicochemical properties of copper, earth-abundant and inexpensive, the copper compounds are one of the most representative catalysts in both organic transformations. However, challenges of existing catalytic systems still remain, especially copper catalytic systems: Lack of the catalytic system that could catalyze efficiently and regioselectively both the A³ coupling reaction and the redox-A³ reaction with broad substrates scope of aliphatic aldehydes, alkynes and amines under mild conditions. Efficient and practical new copper-basic material catalysts to achieve both reactions at room temperature are highly desirable.

Metal nanoclusters (NCs) are highly representative nanostructured material catalysts that are widely used in various catalytic transformations, such as chemical preparation, energy catalysis and electrochemical catalysis[28–31]. In contrast to metal nanoparticles (NPs), ligand-protected NCs with well-defined structures, high surface-to-volume ratios, and molecular purity can provide active sites with

[1]Henan Key Laboratory of Crystalline Molecular Functional Materials, Henan International Joint Laboratory of Tumor Theranostcal Cluster Materials, Green Catalysis Center, and College of Chemistry, Zhengzhou University, Zhengzhou, P. R. China. [2]College of Chemistry and Chemical Engineering, Henan Polytechnic University, Jiaozuo, P. R. China. ✉e-mail: zangsqzg@zzu.edu.cn

**Fig. 1 | Representative synthesis strategies of propargylamines and C1-propargylamines. a** The traditional method for synthesizing propargylamine. **b** The transition-metal catalyzed the A$^3$ coupling reaction. **c** The transition-metal catalyzed the redox-A$^3$ coupling reaction. **d** This work: The Cu$_3$NC$^{(NHC)}$ catalyzed both the A$^3$ coupling reaction and the redox-A$^3$ coupling reaction. R$_1$, R$_2$, R$_3$ and R represent kinds of functional group.

relatively uniform distributions, facilitate activation of reactants, allow techniques for activity and selectivity control in certain cases, and reveal the correlation between the structure and catalytic performance at the atomic scale[32–34]. Importantly, unlike organometallic catalysts, NCs catalysts with metal-metal interactions are conducive to achieving the synergistic catalysis effect of multiple metals in the special organic transformations, potentially improving practical properties of the catalysts[32–35]. However, the exploration of copper NCs is more uncommon than that of other coin metal NCs. The long-term stability of copper NCs in organic transformation systems may remain an unrealized problem. The versatile ligands of N-heterocyclic carbenes (NHCs) are widely used in diverse transition metal catalysts for catalytic organic transformations[36–39]. Due to the high modularity of NHC structures, the unique steric and electronic effects and formation of stable C–metal bonds between the NHC ligands and metal atom, NHC ligands are ideal candidates for improving the stability and reactivity of copper NC catalysts through ligand engineering effects[39–41]. Copper NCs have not been investigated as the model catalysts to realize highly all-in-one dehydrogenative coupling reactions at room temperature.

Herein, we design and develop a tridentate N-heterocyclic carbene that serves as the protected ligand of copper NCs, and a rigid-flexible-coupled copper cluster [Cu$_3$(NHC)$_3$(PF$_6$)$_3$] (Cu$_3$NC$^{(NHC)}$) is constructed successfully in the gram scale. The designed N-heterocyclic carbene ligand endows Cu$_3$NC$^{(NHC)}$ with dual attributes of rigidity and flexibility. Due to the formation of the stable C–Cu bonds and the N–Cu coordination bonds between the NHC ligand shell with the dynamic L-ligand (DLL, L: NHC) and the metal core, this ligand can improve the Cu$_3$NC$^{(NHC)}$ rigidity, which is beneficial for further enhancing the stability of the Cu$_3$NC$^{(NHC)}$. The pyridine of the N-heterocyclic carbene is designed as DLL, which favors a dynamic balance between the pyridine and aliphatic amines to protect catalytic centers and prevent Cu$_3$NC$^{(NHC)}$ deactivation. The dissociation and coordination of DLL endow Cu$_3$NC$^{(NHC)}$ with flexible features, further improving regioselective dehydrogenative coupling reactions through weak interactions between dissociated pyridines and reactants. Importantly, the Cu$_3$NC$^{(NHC)}$ catalyst is applied to achieve efficient and regioselective all-in-one three-component dehydrogenative coupling reactions with inert substrates at room temperature (Fig. 1d). By combining single-crystal structure analyses, experimental characterizations, and DFT calculations, we find the highly regioselective dehydrogenative coupling mechanisms and the relationships between structures and catalytic performance characteristics.

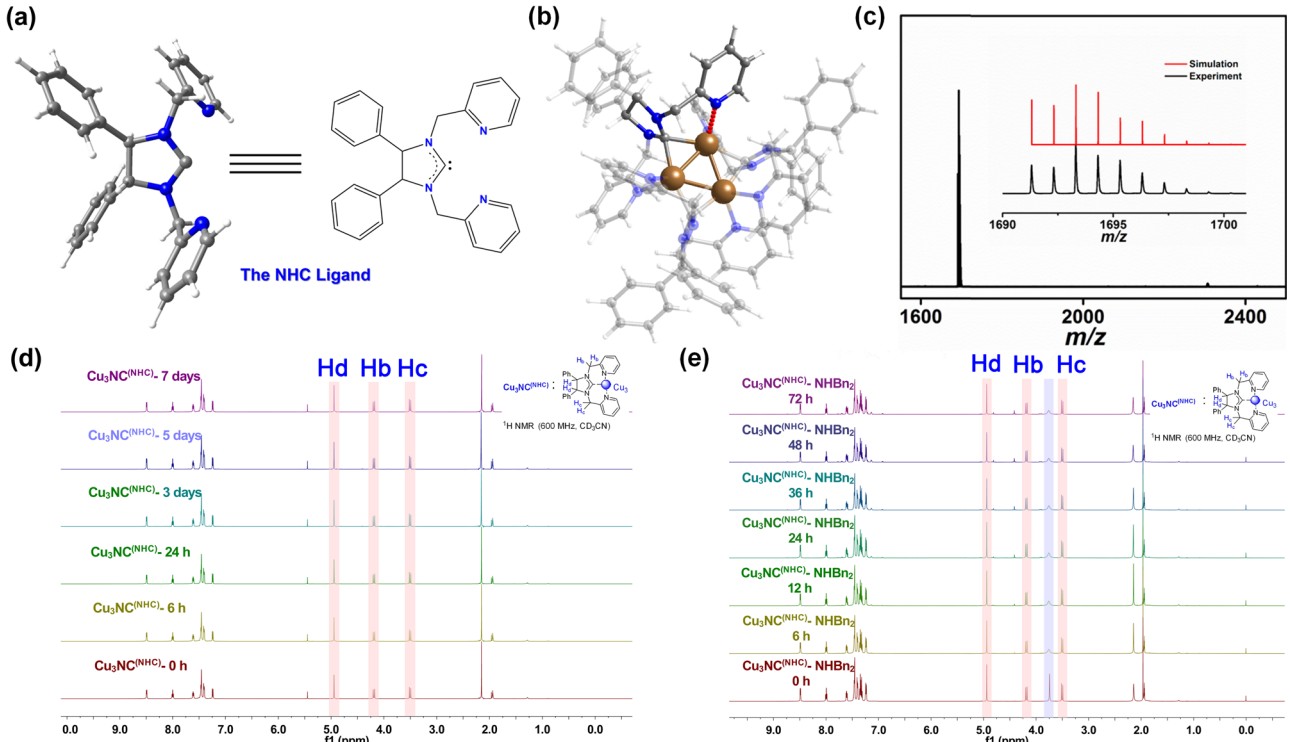

**Fig. 2 | Schematic illustration and characterization of Cu₃NC^(NHC). a** Structure of the N-heterocyclic carbene (NHC) ligand. Color legend: blue, N; gray, C; and white, H. **b** Total structure of the Cu₃NC^(NHC). Color legend: brown, Cu; blue, N; gray, C; and white, H. **c** ESI–MS spectra of Cu₃NC^(NHC) dissolved in dimethylformamide (DMF) /DCM solution and measured in the positive mode. Inset: experimental (black) and simulated (red) isotopic distributions of Cu₃NC^(NHC). **d** ¹H NMR spectra of Cu₃NC^(NHC) in MeCN-d₃ (0–7 days). Hd, Hb and Hc represent the characteristic H atoms of Cu₃NC^(NHC). **e** ¹H NMR spectra of Cu₃NC^(NHC)-HNBn₂ (1:3) in MeCN-d₃ (0–72 h). Hd, Hb and Hc represent the characteristic H atoms of the Cu₃NC^(NHC). The characteristic peaks in the pale blue box represent the methylene peak of the HNBn₂.

## Results

### Catalyst development and characterization

The N-heterocyclic imidazolium salt 4,5-diphenyl-1,3-bis(pyridin-2-ylmethyl)−4,5-dihydro-1H-imidazol-3-ium chloride was designed and synthesized by the condensation reaction and the nucleophilic substitution transformation (Supplementary Fig. 1), forming a model tridentate NHC ligand including a 5-membered N-heterocyclic carbene and two pyridines of DLL in NCs (Fig. 2a). Cu₃NC^(NHC) was prepared using the designed tridentate NHC ligand and copper powder as the copper source. The [Cu₃(BINAP)₃CO₃](ᵗBuSO₃) (Cu₃NC^(BINAP)) and the Cu₃(3,5-Ph₂-Pz)₃ (Cu₃NC^(Pz)) were synthesized by employing 2,2-bis(diphenylphosphino)−1,1-binaphthyl (BINAP) and 3,5-diphenyl-pyrazole (Pz) ligands, respectively, through solvent diffusion methods[42,43]. More synthetic details can be found in the Supplementary Information. Single-crystal X-ray analysis revealed that Cu₃NC^(NHC) contained a triangle Cu₃ metal kernel that was stabilized by three NHC ligands, in which three PF₆⁻ anions balanced the charges (Fig. 2b). Due to the perpendicular orientation of the carbene ligands to the Cu₃ face in Cu₃NC^(NHC), the pyridine rings of the NHC ligands could alternate their coordination above and below the Cu₃ plane. Of the three NHC ligands, all ligands bonded to the Cu₃ core in the same coordination manner via Cu–C bonds and Cu–N bonds, which improved the Cu₃NC^(NHC) rigidity and the stability of Cu₃NC^(NHC). The flexibility of Cu₃NC^(NHC) originates in the flexible bonding and dissociation between pyridine groups of ligands and Cu centers. When the original pyridine moieties in DLL ligands were dissociated from Cu₃ core in the catalytic systems, pyridines could rotate freely to decrease the steric hindrance around the copper center, making the catalytic center exposed to the catalytic substrate. While the reaction was completed, pyridine groups reversibly coordinated to Cu atoms, restoring the original structure of the cluster. The flexibility ensures the stability and reversibility of this copper cluster catalyst. The Cu···Cu distances of Cu₃NC^(NHC) ranged

from 2.4960 to 2.5238 Å with an average distance of 2.5124 Å; this value is shorter than the sum of the van der Waals radii of copper (2.8 Å), suggesting the presence of strong metal-metal interactions[44,45]. To ascertain the exact molecular masses and formulas of the clusters, Cu₃NC^(NHC) and Cu₃NC^(BINAP) were further characterized by electrospray ionization mass spectrometry (ESI–MS) in positive mode (Fig. 2c, Supplementary Figs. 7 and 8). For Cu₃NC^(NHC), the spectrum depicted prominent peaks at m/z 1693.316, corresponding to the molecular ion of [Cu₃(NHC)₃(PF₆)₂]⁺ (calcd m/z 1693.315) (Fig. 2c). For Cu₃NC^(BINAP), the peak with high abundance at m/z 2118.375 was assigned to [Cu₃(BINAP)₃CO₃]⁺ (calcd m/z 2118.375) (Supplementary Fig. 8). The molecular ion peaks of Cu₃NC^(NHC) and Cu₃NC^(BINAP) were in precise agreement between experimental and simulated isotopic distributions.

The phase purities of Cu₃NC^(NHC), Cu₃NC^(BINAP) and Cu₃NC^(Pz) were confirmed by powder X-ray diffraction measurements (Supplementary Figs. 4–6). In addition, the nuclear magnetic resonance (NMR) spectra of these clusters were characterized (Supplementary Figs. 9–14). According to the in situ ¹H NMR spectra of Cu₃NC^(NHC) in CD₃CN or basic solvent, the characteristic peaks of Cu₃NC^(NHC) were not shifted for 7 days or 72 h, confirming that Cu₃NC^(NHC) is very stable in organic solvents (Fig. 2d, e). Supplementary Fig. 15 displayed the ultraviolet–visible (UV–vis) absorption spectrum of Cu₃NC^(NHC) in dichloromethane (DCM), with three characteristic peaks at 290 nm, 346 nm and 389 nm. In addition, time-dependent UV–vis spectra showed no discernible change in DCM or basic solvent (Supplementary Figs. 16 and 17). The UV–vis absorption behavior of Cu₃NC^(BINAP) in DCM or basic solvent was unchanged after 72 h of treatment (Supplementary Figs. 18 and 19). The time-dependent UV–vis spectra of Cu₃NC^(Pz) showed no change in DCM (Supplementary Fig. 20). The in situ ¹H NMR spectra of the Cu₃NC^(Pz) characteristic peaks in CD₃Cl or basic solvent were not shifted after 148 h and 72 h of treatment,

**Table 1 | Optimization of $A^3$ coupling reaction conditions[a,b]**

| Entry | Variations | Conv. | Yield of 4 |
|---|---|---|---|
| 1 | None | 100% | 99% |
| 2 | 2 h | 65% | 65% |
| 3 | 5 h | 100% | 99% |
| 4 | $Cu_3NC^{(BINAP)}$, 5 h | 70% | 68% |
| 5 | $Cu_3NC^{(Pz)}$, 5 h | 27% | 25% |
| 6 | 5 mol% CuBr, 5 h | 50% | 48% |
| 7 | 5 mol% CuCl, 5 h | 28% | 27% |
| 8 | 5 mol% $Cu_2O$, 5 h | trace | trace |
| 9 | $Cu(MeCN)_4PF_6$, 5 h | 61% | 59% |
| 10 | Dioxane as solvent | 20% | <10% |
| 11 | THF as solvent | 30% | 15% |
| 12 | MeOH as solvent | n.r | n.r |
| 13 | MeCN as solvent | 70% | 53% |
| 14 | No copper | n.r. | n.r. |
| 15 | NHC ligand | n.r. | n.r. |
| 16 | No 4 Å MS | 20% | 20% |

[a]General reaction conditions: 1 (0.2 mmol, 1.0 equiv.), 2 (0.2 mmol, 1.0 equiv.), 3 (0.3 mmol, 1.5 equiv.), [Cu] catal. (0.005 mmol, 2.5 mol%), activated MS 4 Å sieves (600 mg) in dry solvent (2.0 mL) under $N_2$ at room temperature.
[b]Conversion and yield were determined by $^1H$ NMR using 1,3,5-trimethoxybenzene as an internal standard.

respectively (Supplementary Figs. 24 and 25). All the characterization of the clusters indicated that $Cu_3NC^{(NHC)}$, $Cu_3NC^{(BINAP)}$ and $Cu_3NC^{(Pz)}$ were successfully synthesized, and their structures were retained intact in organic solvents.

## $A^3$ coupling and redox-$A^3$ coupling reaction development

By investigating the structural and chemical characteristics of $Cu_3NC^{(NHC)}$, we tested the material performance in both organic transformations. Initially, by using 2.5 mol% $Cu_3NC^{(NHC)}$ as the catalyst, benzaldehyde 1 (0.2 mmol, 1.0 eq.), phenylacetylene 2 (0.2 mmol, 1.0 eq.), molecular sieves (MS) 4 Å sieves, dibenzylamine 3 (0.3 mmol, 1.5 eq.) and dry DCM (2 mL) were added to dry Schlenk tube in proper order and stirred under $N_2$ atmosphere at room temperature for 2 h to examine the catalytic activity of $Cu_3NC^{(NHC)}$ in the $A^3$ coupling reaction (Table 1, entry 2). Surprisingly, $Cu_3NC^{(NHC)}$, as a new type catalyst, catalyzed the $A^3$ coupling reaction, affording the desired typical propargylamine product with good yields (65%). Previously, it was reported that Au NPs and metal nanoclusters were utilized as catalysts to catalyze the $A^3$ coupling reaction under harsh conditions by Vinu and Jin's group et al.[46–54]. Strikingly, the $Cu_3NC^{(NHC)}$ catalyst could efficiently catalyze the $A^3$ coupling reaction at room temperature. Inspired by this result, we set the goal to explore the possibility of the $Cu_3NC^{(NHC)}$ as a catalyst, which was employed in the $A^3$ coupling reaction. The performance attributes of some alternative conditions are summarized in Table 1.

As shown in Table 1, an excellent yield (99%) of the desired propargylamine product was obtained by prolonging the reaction time from 2 to 5 h (Table 1, entries 2 and 3). Under the same reaction conditions, $Cu_3NC^{(BINAP)}$ and $Cu_3NC^{(Pz)}$ as catalysts could catalyze the $A^3$ coupling reaction, providing propargylamine in good yields (68%) and poor yields (25%) at room temperature for 5 h, respectively (Table 1, entries 4 and 5). For the $A^3$ coupling reaction, $Cu_3NC^{(NHC)}$

exhibited higher catalytic activities than $Cu_3NC^{(BINAP)}$ and $Cu_3NC^{(Pz)}$. Interestingly, propargylamine with excellent yield (99%) was still obtained (Table 1, entry 1) when $Cu_3NC^{(NHC)}$ was utilized as the catalyst in the $A^3$ coupling reaction and the reaction time was prolonged from 5 to 12 h. The results showed that $Cu_3NC^{(NHC)}$ was a mild catalyst for propargylamine at room temperature. Notably, the load of the $Cu_3NC^{(NHC)}$ catalyst could be reduced to 0.025 mol% such that the $A^3$ coupling transformation reacted in a higher turnover number (TON) of 3800 than other catalysts at room temperature (Supplementary Table 1)[46,55,56].

As observed, the 5 mol% CuBr, 5 mol% CuCl, 5 mol% $Cu_2O$ and $Cu(MeCN)_4PF_6$ catalysts only had poor to modest yields after 5 h (48%, 27%, trace and 59%, respectively; Table 1, entries 6–9). The other solvents (dioxane, tetrahydrofuran (THF), MeCN and MeOH) were not suitable with a seriously decreased yield (Table 1, entries 10–13). The control experiments showed that activated MS 4 Å sieves alone or the NHC ligand as a catalyst could not catalyze the reaction (Table 1, entries 14 and 15). In addition, the lack of activated MS 4 Å sieves was not conducive to obtaining a great yield of the desired propargylamine product (Table 1, entry 16). In the three-component dehydrogenative coupling reaction systems, the activated MS 4 Å sieves, as a kind of adsorption drying agent, could remove by-product water in time, enhancing dehydrogenative coupling reactions rate and improving the yields of propargylamines to 99%. Moreover, $Cu_3NC^{(NHC)}$ remained intact in the organic catalytic system when serving as the catalyst, as practically no apparent changes in the UV–vis spectra, the in situ $^1H$ NMR and ESI–MS monitoring of organic reaction process were found before and after the transformation (Supplementary Figs. 26–28). These results indicated that multiple C−Cu bonds and N−Cu coordination bonds between the NHC ligands and $Cu_3$ core could enhance the stability of the $Cu_3NC^{(NHC)}$ by way of improving the rigidity of $Cu_3NC^{(NHC)}$.

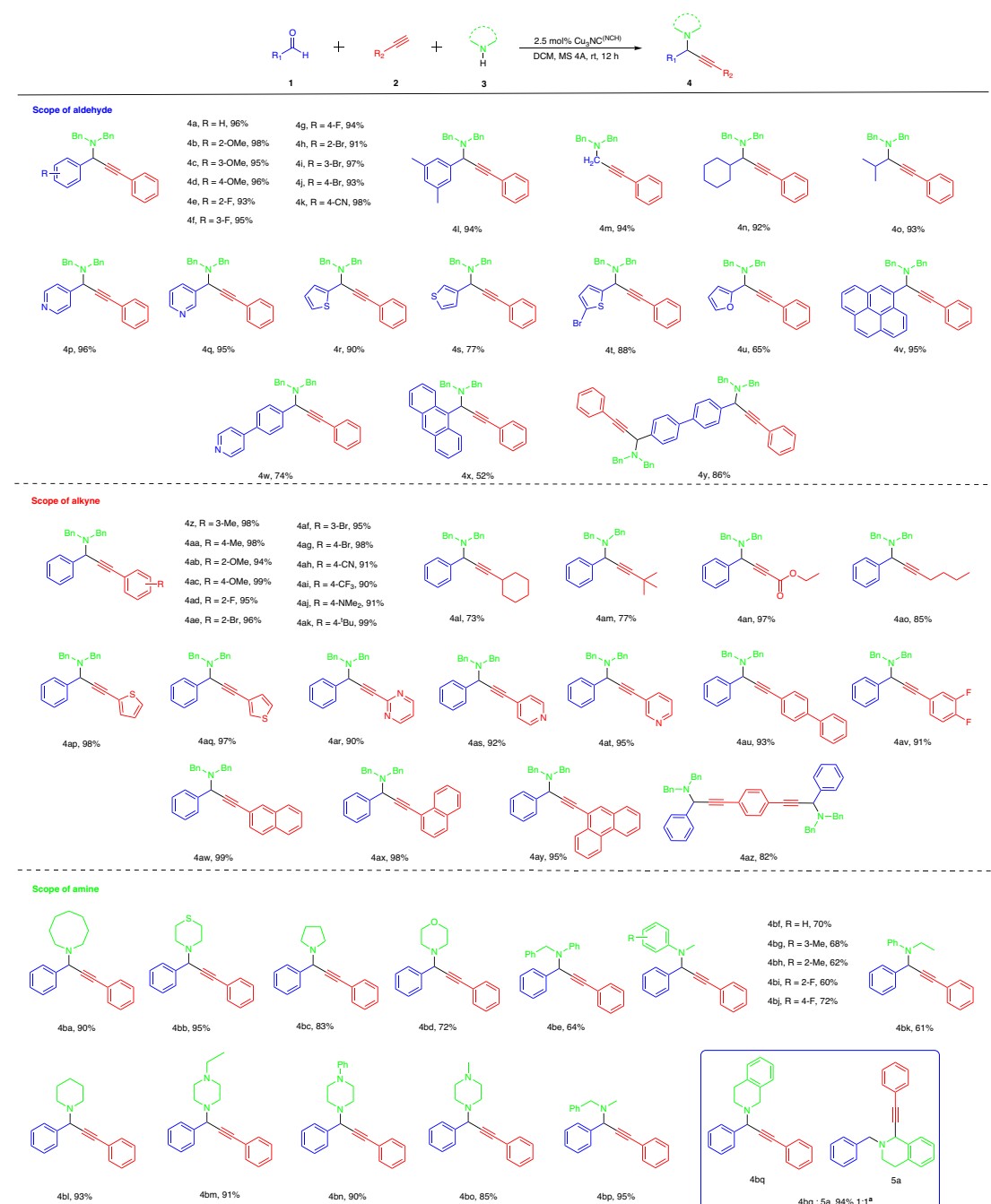

**Fig. 3 | The A³ coupling reaction scope for aldehydes, alkynes and amines.**
General reaction conditions: 1 (0.2 mmol, 1.0 equiv.), 2 (0.2 mmol, 1.0 equiv.), 3 (0.3 mmol, 1.5 equiv.), Cu₃NC(NHC) (0.005 mmol, 2.5 mol%), activated MS 4 Å sieves (600 mg) in dry solvent (2.0 mL) under N₂ at room temperature. Isolated yields are given. ᵃThe ratio of **4bp** and **5a** was determined by crude ¹H NMR spectra of the reaction mixture.

Under the optimal conditions, the substrate scopes of the A³ coupling reaction, including aldehydes, alkynes, and amines, were evaluated. The results are summarized in Fig. 3. We first evaluated the scope of aldehydes. Various aldehydes were prepared as coupling companions with alkynes and amines. For aryl aldehydes (**4a–4l**), the reaction was applicable to both electro-deficient and electron-rich aromatic rings, providing the desired propargylamine products in excellent yields. Aliphatic aldehydes afforded the desired A³ coupling products with excellent yields (**4m–4o**). Notably, electron-deficient heterocyclic pyridine aldehydes, the electron-rich heterocyclic thiophene aldehydes and furan aldehydes were suitable substrates for this transformation with good to excellent yields (**4p–4u**). In addition to

the abovementioned aldehydes, aryl aldehydes containing large substituent groups were explored; these aryl aldehydes performed well, providing the desired products in modest to excellent yields (**4v–4x**). Meanwhile, aryl dialdehyde was a very suitable substrate for this reaction with excellent yield (**4y**).

Various alkynes, including aliphatic and aryl alkynes, were applied to this transformation. Similar to aldehyde substrates, Cu₃NC(NHC) as the catalyst was widely applicable to various alkynes. Both aryl alkynes (**4z–4ak, 4av**) and aliphatic alkynes (**4al–4ao**), including electro-deficient and electron-rich functional groups, displayed good to excellent yields. Both S-heterocyclic and N-heterocyclic aryl alkynes were suitable substrates for this transformation and afforded excellent

yields (**4ap**–**4at**). Furthermore, aryl alkynes containing large substituent groups and aryl dialkyne were investigated and these aryl alkynes worked well, affording the corresponding products in excellent yields (**4au, 4aw**–**4az**) and further supporting our hypothesis of $Cu_3NC^{(NHC)}$ with the flexible feature that the dissociation of Cu−N bonds between pyridine groups and $Cu_3$ core allowed pyridines to rotate freely through the C-C single bond, thereby decreasing the steric hindrance around copper catalytic center and enhanced the catalytic activity.

In addition, a series of amines were explored in the reaction. To our satisfaction, the N-heterocyclic secondary amines were applicable to this transformation with great to excellent yields at room temperature (**4ba, 4bc and 4bl**). Both piperazines with different substituent groups and morpholines (including thiomorpholine) displayed high reactivities and afforded good to excellent yields (**4bb, 4bd, 4bm**–**4bo**). These results highlighted that the pyridine of the DLL was crucial for the $Cu_3NC^{(NHC)}$ catalyst to present a dynamic balance between the DLL and aliphatic amines, maintaining the efficient catalytic activity. Undoubtedly, acyclic secondary amines with the electron-deficient and electron-rich fundamental groups readily afforded propargylamines in good to excellent yields (**4be**–**4bk, 4 bp**). The primary amines, aniline, p-toluidine, 4-fluoroaniline, amantadine, thiophene-2-ethylamine, and 1-octylamine, did not offer propargylamine products, which are considered to be inert substrates for $A^3$-coupling transformations under mild conditions[57,58]. Interestingly, when 1,2,3,4-tetrahydroisoquinoline (THIQs) was used to this transformation, the products of propargylamine and C1-propargylamine were obtained in excellent yields and modest regioselectivity (**4bq and 5a**). Strikingly, the ratio of product **4bq** to product **5a** was 1:1, as determined by $^1H$ NMR. This finding demonstrated that the unique physicochemical properties of the designed $Cu_3NC^{(NHC)}$ catalyst, including the weak interactions between dissociated DLL and reactants, led to this result.

To our knowledge, no one catalytic system could catalyze both the $A^3$ coupling reaction and the redox-$A^3$ reaction at room temperature in an efficient and regioselective manner to date. By considering the catalytic diversity of $Cu_3NC^{(NHC)}$, the $Cu_3NC^{(NHC)}$ was investigated as a new typical catalyst for the synthesis of the desired C1-propargylamine products. As seen in Table 2, with decreasing catalyst loading of $Cu_3NC^{(NHC)}$ from 2.5 to 0.25 mol% and increasing reaction time from 12 to 40 h, the yields of the C1-propargylamine product increased from 48% to 98%, and the regioselectivity gradually increased to 100% at room temperature (Table 2, entries 1–5). By employing toluene as the solvent, the reaction afforded 100% regioselectivity, but the yield was poor (Table 2, entries 6 and 7). To improve the yields of C1-propargylamine, a mixed solvents of toluene-DCM was used in the catalytic system with a slightly improved yield, but the regioselectivity was seriously decreased (Table 2, entries 8 and 9). Notably, the load of the $Cu_3NC^{(NHC)}$ catalyst could be lowered to 0.025 mol% such that the redox-$A^3$ coupling transformation reacted in a high TON of 1840 (Supplementary Table 2). For comparison, the $Cu_3NC^{(BINAP)}$, $Cu_3NC^{(Pz)}$ and copper salts as catalysts were investigated. Under the optimal reaction conditions, $Cu_3NC^{(BINAP)}$ and $Cu_3NC^{(Pz)}$ as catalysts could catalyze the redox-$A^3$ coupling reaction giving C1-propargylamines with moderate to poor yields (50%, 25%) and poor to great the regioselectivity (**4bp:5a**, 1:5 and 0:1) at room temperature (Table 2, entries 10 and 11). As observed, the CuI and $Cu(MeCN)_4PF_6$ catalysts, which were 5 times the catalyst loading of the $Cu_3NC^{(NHC)}$, had modest yields and poor regioselectivity (Table 2, entries 12 and 13). The control experiment showed that neither activated MS 4 Å sieves nor the NHC ligand as a catalyst could catalyze the reaction (Table 2, entries 14 and 15). In addition, the $Cu_3NC^{(NHC)}$ catalyst showed excellent stability in the organic catalytic system, as demonstrated by ESI–MS spectra monitoring before and after the transformation (Supplementary Fig. 29).

Under the optimized conditions, the substrate scope of the redox-$A^3$ coupling reaction, including aldehydes and alkynes, was evaluated (Fig. 4). First, the scope of aldehydes was evaluated. A range of aldehydes were prepared as coupling companions with alkynes and THIQs. For aryl aldehydes (**5a–5d**), the reaction performed well for both electro-deficient and electron-rich aromatic rings, offering the desired C1-propargylamine products in high yields and good to excellent regioselectivity. Furthermore, the electron-rich heterocyclic thiophene aldehydes, aryl aldehydes containing large substituent groups and aliphatic aldehyde afforded the desired and single redox-$A^3$ coupling product with excellent yields (**5e–5 g**). Next, various alkynes were explored. Both aryl alkynes (**5h–5k**) and aliphatic alkynes (**5n**) with electro-deficient and electron-rich functional groups were well applied to the reaction in high yields and excellent regioselectivity. Interestingly, both aryl alkynes containing large substituent groups and S-heterocyclic aryl alkynes were tolerated in the $Cu_3NC^{(NHC)}$-catalyzed redox-$A^3$ coupling reaction, giving single and excellent yields of C1-propargylamines (**5l and 5m**). Finally, the other secondary amines, N-phenylbenzylamine (**3be**), 1-ethylpiperazine (**3bm**), 1-phenylpiperazine (**3bn**), N-methylbenzylamine (**3bp**), and pyrrolidine (**3bc**), were utilized to the redox-$A^3$ coupling transformation under optimized conditions. Regretfully, the redox-$A^3$ coupling products were not observed, indicating that in situ-generated exo-iminium ions are not always easy to isomerize to *endo*-iminium ions.

## Mechanistic studies and density functional theory (DFT) calculations

To clarify why $Cu_3NC^{(NHC)}$ could catalyze highly all-in-one three-component dehydrogenative coupling reactions and to gain insight into the catalytic mechanism of these transformations, DFT calculations and a series of control experiments were conducted. First, the characteristic peaks of the benzyl-methylene proton of $NHBn_2$ began to broaden in the in situ $^1H$ NMR spectra of the mixture the $Cu_3NC^{(NHC)}$ and $NHBn_2$ with increasing reaction time, and the characteristic peaks of $Cu_3NC^{(NHC)}$ were slightly shifted (Fig. 2e, Supplementary Figs. 22 and 23). The results illustrated that $Cu_3NC^{(NHC)}$, which was coordinated with the N atom of $NHBn_2$, had good adsorption ability on amine reactants. Moreover, we considered the bonding and adsorption characteristics of alkynyl substrates on the surface active site of $Cu_3NC^{(NHC)}$ by using 2-fluorophenylacetylene (2-F-PA) as an example, which was convenient for $^1H$ and $^{19}F$ NMR to follow the transformations process. Experimentally, obvious chemical shifts were revealed for the in situ $^1H$ and $^{19}F$ NMR spectra of the 2-F-PA after mixing with the $Cu_3NC^{(NHC)}$, and the characteristic peaks of the catalyst were slightly shifted in the in situ $^1H$ NMR and $^{19}F$ NMR spectra, demonstrating a non-negligible interaction between the acetylene of 2-F-PA and $Cu_3NC^{(NHC)}$ (Supplementary Figs. 31 and 32). Next, we explored the transformations process of the $A^3$ coupling reaction and redox-$A^3$ coupling reaction by taking the in situ $^{19}F$ NMR spectra, suggesting that the characteristic peaks of 2-F-PA started to gradually disappear at −110.10 ppm and the characteristic peaks of the propargylamines and C1-propargylamines began to appear at −109.42 ppm and −109.70 ppm, respectively (Supplementary Figs. 33 and 34), indicating the $Cu_3NC^{(NHC)}$ catalyst possessed the excellent regioselectivity in the $A^3$ coupling reaction and redox-$A^3$ coupling reaction. Finally, to obtain convincing evidence for the $Cu_3NC^{(NHC)}$-activated deprotonation of alkynes to form $R_1C≡CCu_3NC$, the control experiments were designed to prove the mechanism of $Cu_3NC^{(NHC)}$-activated deprotonation of alkynes, in which the PhC≡CD instead of PhC≡CH was used in the $A_3$ coupling reaction and redox-$A_3$ coupling reaction. The characteristic peaks of D were not present in the $^2H$ NMR spectra of the purified propargylamines and C1-propargylamines (Supplementary Fig. 35), revealing that the terminal D fell off and formed $R_1C≡CCu_3NC$ during the $A^3$ coupling reaction and redox-$A^3$ coupling reaction.

**Table 2 | Optimization of redox-A³ coupling reaction conditions[a,b]**

| Entry | Variations | Conv. | Yield of 4 bp | Yield of 5a | Ratio |
|---|---|---|---|---|---|
| 1 | None | 100% | — | 98% | 0:1 |
| 2 | 2.5 mol% Cu$_3$NC$^{(NHC)}$, 12 h | 100% | 47% | 47% | 1:1 |
| 3 | 1.0 mol% Cu$_3$NC$^{(NHC)}$, 12 h | 82% | 17% | 63% | 1:3.7 |
| 4 | 0.5 mol% Cu$_3$NC$^{(NHC)}$, 20 h | 78% | 11% | 66% | 1:6 |
| 5 | 0.25 mol% Cu$_3$NC$^{(NHC)}$, 32 h | 82% | — | 82% | 0:1 |
| 6 | 2.5 mol% Cu$_3$NC$^{(NHC)}$, Toluene, 12 h | 16% | — | 14% | 0:1 |
| 7 | 2.5 mol% Cu$_3$NC$^{(NHC)}$, Toluene, 50 °C, 6 h | 34% | — | 30% | 0:1 |
| 8 | 2.5 mol% Cu$_3$NC$^{(NHC)}$, Toluene-DCM 1:1, 12 h | 38% | 2% | 35% | 1:17.5 |
| 9 | 2.5 mol% Cu$_3$NC$^{(NHC)}$, Toluene-DCM 1:4, 12 h | 70% | 20% | 50% | 2:5 |
| 10 | 0.25 mol% Cu$_3$NC$^{(BINAP)}$ | 62% | 10% | 50% | 1:5 |
| 11 | 0.25 mol% Cu$_3$NC$^{(Pz)}$ | 27% | — | 25% | 0:1 |
| 12 | 1.25 mol% CuI | 66% | 10% | 56% | 1:5.6 |
| 13 | 1.25 mol% Cu(MeCN)$_4$PF$_6$ | 30% | — | 26% | 0:1 |
| 14 | No copper | n.r | n.r | n.r | — |
| 15 | NHC ligand | n.r | n.r | n.r | — |

[a]General reaction conditions: 1 (0.2 mmol, 1.0 equiv.), 2 (0.2 mmol, 1.0 equiv.), 3 (0.3 mmol, 1.5 equiv.), [Cu] catal. (0.0005 mmol, 0.25 mol%), activated MS 4 Å sieves (600 mg) in dry solvent (2.0 mL), under N$_2$ at room temperature.
[b]Conversion, ratio of **4bp** and **5a**, yield of **4bq** and yield of **5a** were determined by $^1$H NMR using 1,3,5-trimethoxybenzene as an internal standard.

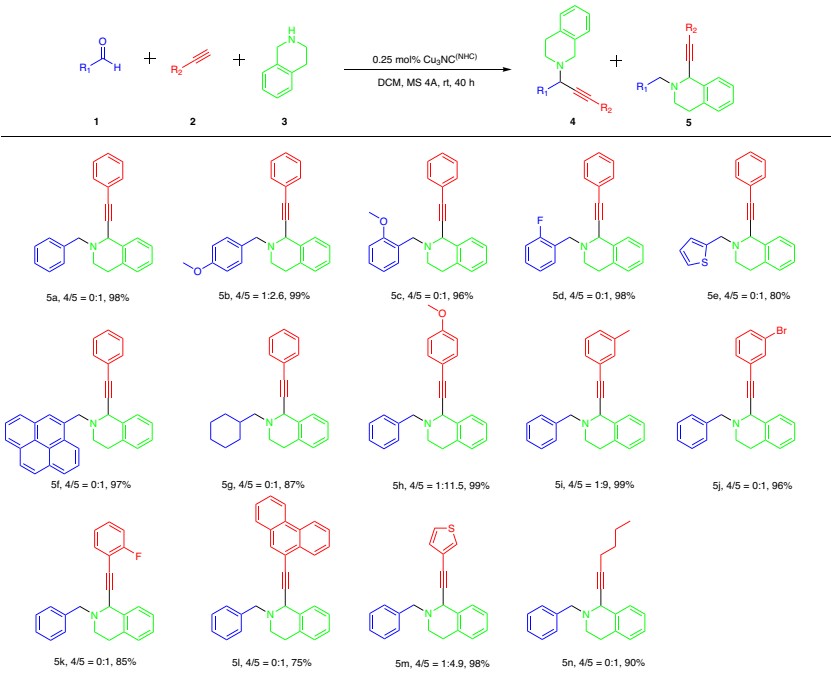

**Fig. 4 | Redox-A³ coupling reaction scope for aldehydes and alkynes.** General reaction conditions: 1 (0.2 mmol, 1.0 equiv.), tetrahydroisoquinoline 2 (0.2 mmol, 1.0 equiv.), 3 (0.3 mmol, 1.5 equiv.), $Cu_3NC^{(NHC)}$ (0.0005 mmol, 0.25 mol%), activated MS 4 Å (600 mg) in dry solvent (2.0 mL) under $N_2$ at room temperature. Isolated yields are given. The ratio of **4** and **5** was determined by the crude ¹H NMR spectra of the reaction mixture.

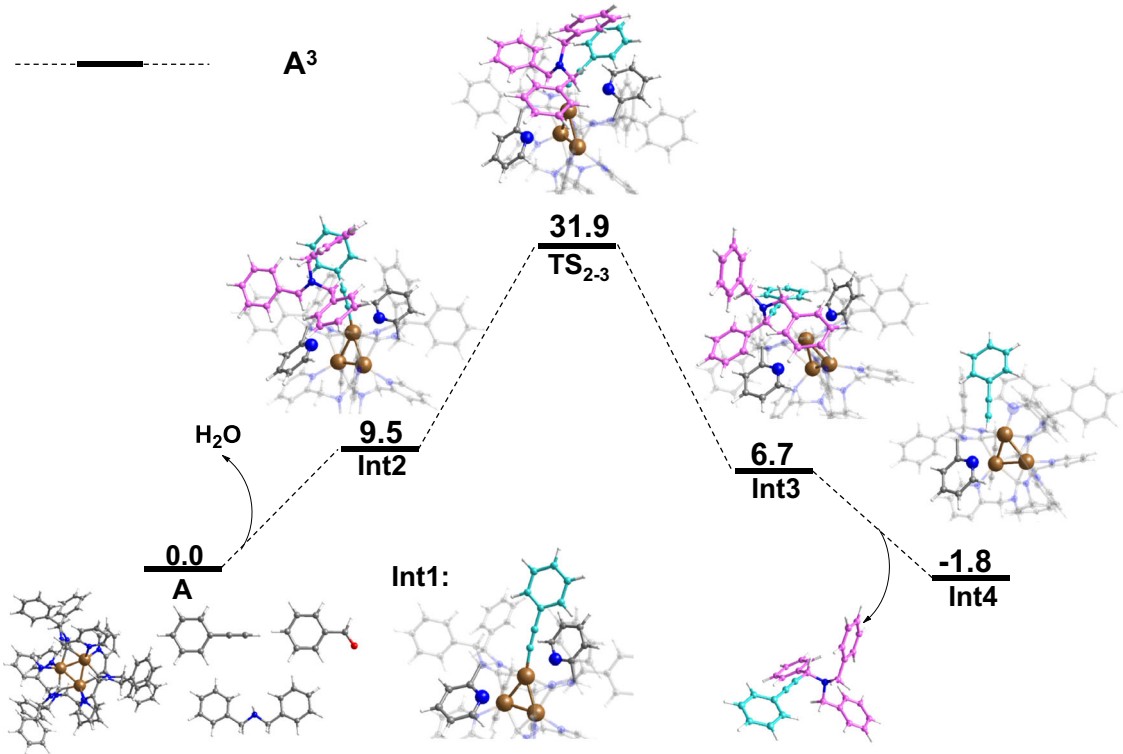

**Fig. 5 | Theoretical calculations.** DFT-computed Gibbs free energy profile ($\Delta G$ in kcal/mol) for the formation of the propargylamine.

To clarify the catalytic mechanism of the A³ coupling reaction, DFT calculations were performed on the specific compounds applied experimentally: benzaldehyde (1), phenylacetylene (2) and dibenzylamine (3) with the $Cu_3NC^{(NHC)}$ catalyst (Fig. 5). The A³ coupling reaction commences when one pyridine of N-heterocyclic carbene could dynamically dissociate from the copper atom of $Cu_3NC^{(NHC)}$ in the organic transformation, exposing the catalytic copper sites of the $Cu_3NC^{(NHC)}$ and activating the phenylacetylene (2) substrate. Then the other pyridine of the N-heterocyclic carbene dynamically dissociates from the single copper atom of $Cu_3NC^{(NHC)}$ to release more

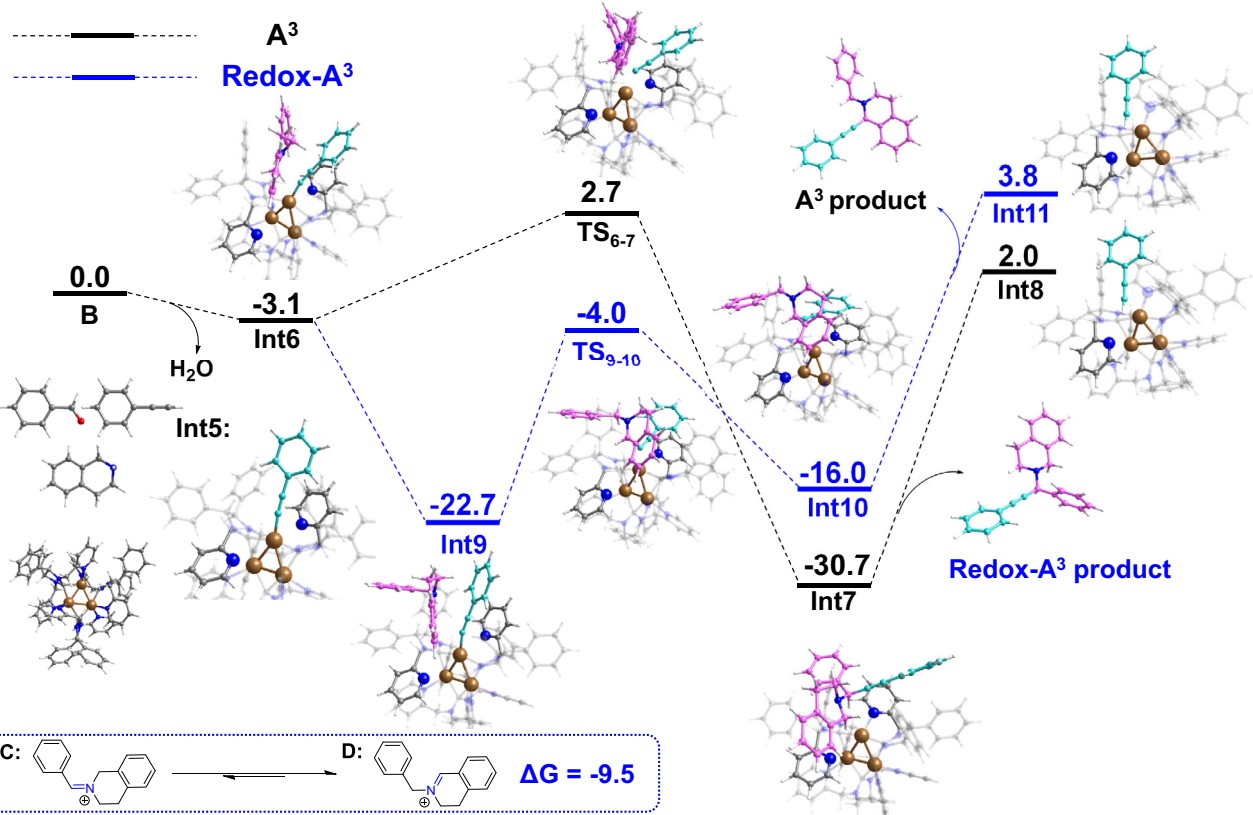

**Fig. 6 | Theoretical calculations.** DFT-computed Gibbs free energy profile (ΔG in kcal/mol) for the formation of the C1-propargylamines.

space and in situ generate copper acetylide under weak bases environment. Notably, the C–Cu bonds between the NHC ligands and $Cu_3$ core have always been very stable during the organic transformation process to keep catalyst integrity. The in situ-generated copper acetylide Int1 and iminium cation B afford the ion pair Int2. This process from initial state A to ion pair Int2 leads to an energy increase of 9.5 kcal/mol and produces one molecule of water. Subsequently, the reaction of copper acetylide Int1 and iminium cation B could form propargylamine–$Cu_3NC^{(NHC)}$ complex Int3 via the transition state $TS_{2-3}$, which requires an energy barrier of 31.9 kcal/mol. Next one molecule of propargylamine is dissociated from complex Int3. Furthermore, the half of dissociated Cu–N bond is reformed, and another molecule of phenylacetylene (2) is activated by the $Cu_3NC^{(NHC)}$ catalyst with exposed active catalytic sites to obtain the catalyst Int4, which could enter the next cycle.

To gain a deep understanding of the origin of the highly reactive and regioselective redox-$A^3$ coupling reaction, we performed the DFT calculations with THIQs as the amine substrate replacing dibenzylamine in this system (Fig. 6). The condensation of benzaldehyde (1) with THIQs produces *exo*-iminium ion D, and this process of forming the ion pair Int6 leads to an energy decrease of 3.1 kcal/mol and produces one molecule of water. The *exo*-iminium ion D is very easy to convert to a more stable *endo*-iminium ion E through releasing energy of 9.5 kcal/mol. It is very beneficial that the ion pair Int6 converts to a more stable ion pair Int9 by releasing energy of 19.6 kcal/mol. It means that the intermediate ion pair Int9 is a more stable and easier to produce than ion pair Int6. In addition, according to the energetic span model[59,60], the overall barrier of the $A^3$ coupling reaction is 33.4 kcal/mol, corresponding to the energy difference between the first Int7 and the second $TS_{6-7}$, requiring a higher energy barrier than the overall barrier for the redox-$A^3$ coupling reaction (Supplementary Figs. 36 and 37). All these DFT calculation results are greatly helpful to understand the specificity of the $Cu_3NC^{(NHC)}$ catalyst in the redox-$A^3$ coupling reaction.

Therefore, $Cu_3NC^{(NHC)}$ as the catalyst could catalyze the redox-$A^3$ coupling reaction with high efficiency and high regioselectivity under mild conditions when the THIQs were used as the amine substrates.

Based on the abovementioned results of control experiments and DFT calculations, the plausible mechanisms for the formation of propargylamines and C1-propargylamines are shown in Fig. 7, which both commence with one pyridine of DLL dynamically dissociating from the $Cu_3NC^{(NHC)}$ (I) to expose copper catalytic sites, activate the alkyne substrate and afford the intermediate π-metal–alkyne complex (II). The intermediate complex (II) could make the alkyne proton more acidic, which is conducive to the deprotonation of terminal alkyne by the presence of weak bases (amine reactants, *exo*-iminium ions and so on) in the reaction media. The reactions of aldehyde and amine generate in situ *exo*-iminium ions, which then react with the intermediate π-metal–alkyne complex (II). Moreover, the other pyridine of DLL dynamically dissociates from the single copper atom of $Cu_3NC^{(NHC)}$ to release a relatively high amount of space, which is more conducive to in situ generating the copper acetylides and by-product $H_2O$ under weak bases environment. The by-product $H_2O$ could be removed by the activated MS 4 Å sieves. Then the reaction of in situ-generated copper acetylides and *exo*-iminium ions (via oxidative addition/reductive elimination) could form propargylamines (the $A^3$ coupling product) (Fig. 7, Path A). When in situ-generated exo-iminium ions isomerize to *endo*-iminium ions, the *endo*-iminium ions react with the activated intermediate π-metal–alkyne complex (II) to afford the copper acetylides and by-product $H_2O$. The adsorption drying agent of MS 4 Å sieves could remove by-product water in time, enhancing reactions rate. Next, the metal acetylides react with *endo*-iminium ions, forming C1-propargylamines (the redox-$A^3$ coupling product) through oxidative addition/reductive elimination (Fig. 7, Path B). The dissociated Cu–N bonds of $Cu_3NC^{(NHC)}$ are reformed, indicating the concomitant regeneration of $Cu_3NC^{(NHC)}$ (I) in the dehydrogenative coupling transformations; $Cu_3NC^{(NHC)}$ could enter the next cycle.

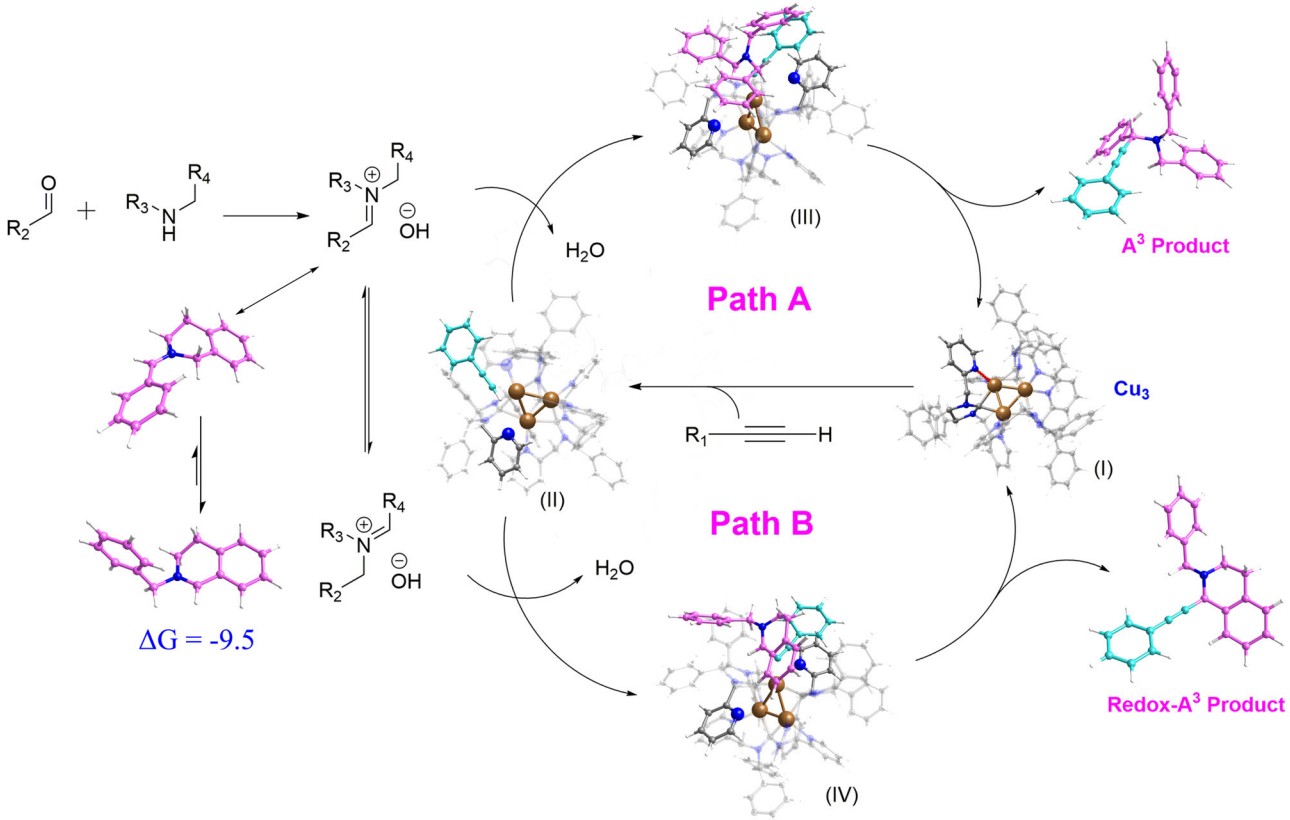

**Fig. 7 | Proposed reaction mechanism (ΔG in kcal/mol). Path A** catalytic mechanism for the formation of the propargylamines. **Path B** catalytic mechanism for the formation of the C1-propargylamines.

To further explore the practical properties of the Cu$_3$NC$^{(NHC)}$ catalyst, preliminary kinetic studies of the A$^3$ coupling reaction and redox-A$^3$ coupling reaction were conducted. Both the reactions followed the pseudo-first-order rate dependence on the concentration of phenylacetylenes, as demonstrated by the linear fitting of the ln(C$_0$/C) vs. reaction time (t) curve (Supplementary Figs. 38–41). Moreover, both large-scale preparations (2.0 mmol) of propargylamines and C1-propargylamines were further studied, affording 95% yield of **4a** and 96% yield of **5a** under mild conditions, respectively (Fig. 8a, b). More importantly, the catalytic system of Cu$_3$NC$^{(NHC)}$ could be used for the synthesis of bioactive molecules, such as N-ethyl-3-carbazolecarboxaldehyde, to provide propargylamine product **4br** in 90% yield under mild conditions (Fig. 8c). Surprisingly, the diastereoselectivity of the Cu$_3$NC$^{(NHC)}$-catalyzed A$^3$ coupling reaction was investigated, in which (R)−1-phenyl-1,2,3,4-tetrahydroisoquinoline as a model substrate presented propargylamine product **4bs** with excellent diastereoselectivity (15.8:1) (Fig. 8d). Finally, the investigation found that the Cu$_3$NC$^{(NHC)}$ catalyst displayed excellent stability and reusability in both transformation reactions. The A$^3$ coupling reaction was conducted six times, and the redox-A$^3$ coupling reaction was conducted three times with the Cu$_3$NC$^{(NHC)}$ catalyst. Nearly quantitative yields were obtained in all transformations (Supplementary Figs. 42 and 43).

## Discussion

In summary, we have successfully developed a rigid-flexible-coupled Cu$_3$NC$^{(NHC)}$ nanocluster in the gram scale, supported by the designed NHC ligand. The atomically precise Cu$_3$NC$^{(NHC)}$ is endowed with a dynamically catalytic center and dual attributes of flexibility and rigidity by dynamic dissociation and coordination of DLL. The ultra-small copper clusters with ultrahigh stability have been used to catalyze highly all-in-one three-component dehydrogenative coupling reactions, including excellent regioselectivity (reaching 100%), high

efficiency (TON = 3800 and 1840), high yield (reaching 99%), broad substrate scopes (85 examples) and mild conditions (room temperature). Mechanistic and control experimental studies have demonstrated that the remarkable catalytic properties originate from the rigid and flexible dual attributes of the Cu$_3$NC$^{(NHC)}$ catalyst. The origin of the enhanced regioselectivity catalyzed by Cu$_3$NC$^{(NHC)}$ is associated with the energy of the ion pair with copper acetylide and iminium cation, leading to modulation of the activation barrier to obtain a single product. This work illustrates the precise structure–activity relationship of nanoclusters and provides a reference for using metal nanoclusters in homogenous organic transformations, which can build a bridge between metal nanoclusters catalyst and organic pharmaceutical chemistry and promote more exploration of metal nanocluster-catalyzed organic reactions.

## Methods

All the reactions were carried out under ambient atmosphere (air) conditions unless otherwise noted. All commercial reagents and solvents were obtained from the commercial provider and used without further purification. $^1$H NMR and $^{13}$C NMR spectra were recorded on Bruker 600 MHz spectrometers. Chemical shifts were reported relative to internal tetramethylsilane (δ 0.00 ppm), CD$_3$CN (δ 1.94 ppm) or CDCl$_3$ (δ 7.26 ppm) for $^1$H NMR and CDCl$_3$ (δ 77.0 ppm), CD$_3$CN (δ 118.3 ppm) for $^{13}$C NMR. Flash column chromatography was performed on 300-400 mesh silica gel.

### Instrumentation

Powder X-ray diffraction (PXRD) patterns of Cu$_3$NC$^{(NHC)}$, Cu$_3$NC$^{(BINAP)}$, and Cu$_3$NC$^{(Pz)}$ were recorded on a Rigaku B/Max-RB X-ray diffractometer with Cu-Kα radiation (λ = 1.5418 Å) in air at room temperature. ESI–MS images of the clusters were recorded on an AB Sciex X500R Q-TOF spectrometer. UV–vis absorption spectra were obtained

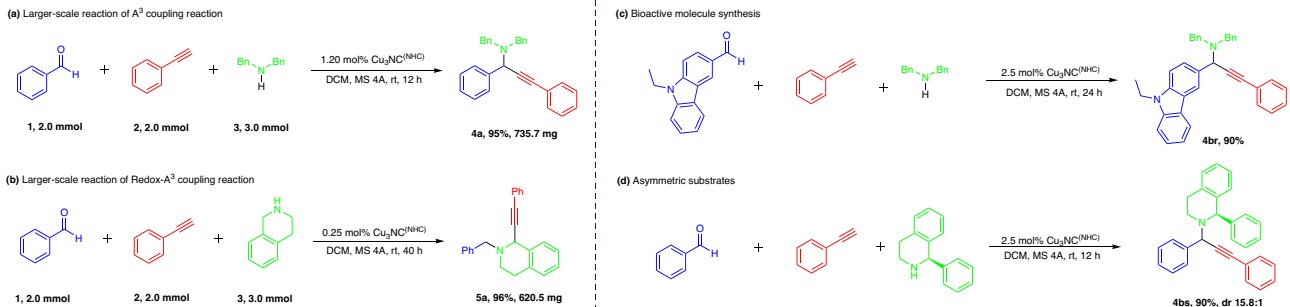

**Fig. 8 | The catalytic performance of Cu₃NC^(NHC). a** Large-scale synthesis of the desired propargylamine product **4a. b** Large-scale synthesis of the desired C1-propargylamine product **5a. c** Bioactive molecule synthesis under mild conditions. **d** Exploration of substrate-controlled asymmetric the Cu₃NC^(NHC)-catalyzed A³ coupling transformation.

through a Hitachi UH4150 UV–visible spectrophotometer. The X-ray diffraction data of Cu₃NC^(NHC) and Cu₃NC^(Pz) were measured on a Rigaku XtaLAB Pro diffractometer (Supplementary Tables 3 and 4).

## Materials

Three-component coupling of aldehydes, alkynes, and amines (including the A³ coupling reaction and the redox-A³ reaction) via C–H activation was carried out under the atmosphere of dried and purified N₂ using standard Schlenk. Toluene and THF were dried over sodium/benzophenone and distilled under nitrogen prior to use. DCM was dried over calcium hydride and distilled under nitrogen prior to use. Ethanol and methanol were dried over Iodine/magnesium strips and distilled under nitrogen prior to use. The other dry solvent including 1,4-dioxane, acetonitrile and DCE were directly purchased from Energy Chemical. BINAP, 3,5-diphenyl-pyrazole, aldehydes, alkynes and amines were purchased from Bidepharm, Energy Chemical and Heowns without further purification. The Vulcan XC-72 carbon black was purchased from Macklin.

## Synthesis of 4,5-diphenyl-4,5-dihydro-1H-imidazole[61]

Under N₂ atmosphere, 1,2-diphenylethylenediamine (3.0 g, 14.1 mmol), dry MeOH (100.0 mL) and formylformic acid (1.5 g, 16.3 mmol) were added into the dry round-bottom flask in turn. The reaction mixture was stirred at room temperature for 4 h. Then NBS (3.0 g, 18.54 mmol) was added into the stirred mixture of the round-bottom flask and the reaction mixture was stirred at room temperature for 20 h. The reaction was monitored by TLC. When 1,2-diphenylethylenediamine was consumed, the reaction was quenched by adding sat. Na₂S₂O₅ (aq.), then concentrated. The residue was added to 5.0% NaOH (aq.) and was extracted by EA. The organic phase was dried by Na₂SO₄, then concentrated. The crude product was then purified by column chromatography to give the 4,5-diphenyl-4,5-dihydro-1H-imidazole as white solid with overall isolated yield 90%. ¹H NMR (600 MHz, CDCl₃) δ 7.36–7.32 (m, 4H), 7.29 (t, *J* = 7.3 Hz, 2H), 7.26–7.23 (m, 4H), 4.70 (s, 2H).

## Synthesis of 4,5-diphenyl-1,3-bis(pyridin-2-ylmethyl)−4,5-dihydro-1H-imidazol-3-ium chloride[62]

Under N₂ atmosphere, 4,5-diphenyl-4,5-dihydro-1H-imidazole (1.1 g, 5.0 mmol), NaHCO₃ (1.3 g, 15.0 mmol), dry EtOH (50.0 mL) and 2-(Chloromethyl) pyridine hydrochloride (1.7 g, 10.3 mmol) were added into the dry round-bottom flask in turn. The reaction mixture was stirred and refluxed at 80 °C by oil bath for 2 days. The reaction was monitored by TLC. When 4,5-diphenyl-4,5-dihydro-1H-imidazole was consumed, the reaction was shut down to cool to ambient temperature, then concentrated. The crude product was then purified by column chromatography to give the 4,5-diphenyl-1,3-bis(pyridin-2-ylmethyl)−4,5-dihydro-1H-imidazol-3-ium chloride as white solid with overall isolated yield 92%. ¹H NMR (600 MHz, CD₃CN) δ 9.44 (s, 1H), 8.69 (d, *J* = 4.3 Hz, 2H), 7.75 (td, *J* = 7.7, 1.7 Hz, 2H), 7.45 − 7.39 (m, 6H), 7.36 (dd, *J* = 7.1, 5.1 Hz, 2H), 7.29 (dd, *J* = 7.7, 1.6 Hz, 4H), 7.23 (d, *J* = 7.7 Hz, 2H), 5.09 (d, *J* = 16.0 Hz, 2H), 5.02 (s, 2H), 4.39 (d, *J* = 16.0 Hz, 2H). ¹³C NMR (151 MHz, CD₃CN) δ 161.5, 153.9, 150.7, 138.4, 136.4, 130.8, 130.4, 129.0, 124.6, 124.0, 74.0, 51.5.

## Synthesis of Cu₃NC^(NHC)

Under N₂ atmosphere, 4,5-diphenyl-1,3-bis(pyridin-2-ylmethyl)−4,5-dihydro-1H-imidazol-3-ium chloride (440.2 mg, 1.0 mmol) was dissolved in 20 mL of H₂O, to which 20 mL of H₂O solution containing KPF₆ (404.8 mg, 2.2 mmol) was added under vigorous stirring at room temperature. The suspension was centrifuged and the white solid was collected. The white solid was dissolved in 30.0 mL of MeCN. Overabundant Cu power (320 mg, 5.0 mmol) was then added to the stirred mixture. The reaction mixture was stirred at room temperature for 24 h. After filtration and concentration, diethyl ether was added to obtain the crude product. The crude product was dissolved in acetonitrile, and the resulting solution was diffused with diethyl ether in vapor phase to obtain colorless crystals of **Cu₃NC^(NHC)** (Yield: 67.8%, calculated based on NHC ligand) for 5 days at room temperature. ¹H NMR (600 MHz, CD₃CN) δ 8.48 (d, *J* = 4.3 Hz, 6H), 7.99 (t, *J* = 7.4 Hz, 6H), 7.63 − 7.56 (m, 6H), 7.51 − 7.28 (m, 30H), 7.23 (d, *J* = 7.5 Hz, 6H), 4.93 (s, 6H), 4.17 (d, *J* = 15.6 Hz, 6H), 3.50 (d, *J* = 15.7 Hz, 6H). ¹³C NMR (151 MHz, CD₃CN) δ 154.2, 152.3, 141.2, 132.2, 131.1, 130.4, 126.4, 126.1, 72.0, 53.0.

## Synthesis of Cu₃NC^(BINAP)[42]

BINAP (0.045 g, 0.07 mmol) and ᵗBuSCu (0.025 g, 0.16 mmol) were well mixed in 15 mL acetonitrile/toluene (volume ratio 2:1) at room temperature. The mixture was treated under ultrasonic conditions until a clear solution was obtained. Then 250 μLCS₂ was added. The solution quickly turned to tawny when exposed to air. The tawny solution was allowed to evaporate slowly in darkness at room temperature. After ~5 days, yellow block crystals of **Cu₃NC^(BINAP)** were obtained in a yield of 49.0% (calculated based on BINAP). ¹H NMR (400 MHz, DMSO) δ 7.98 (s, 12H), 7.59 (d, *J* = 8.1 Hz, 12H), 7.44−7.29 (m, 20H), 7.25 (t, *J* = 7.5 Hz, 5H), 7.15 (dd, *J* = 15.0, 7.1 Hz, 12H), 6.73 (dd, *J* = 34.8, 27.5 Hz, 23H), 6.45 (s, 12H), 1.10 (s, 9H).

## Synthesis of Cu₃NC^(Pz)[43]

Cu(CH₃CN)₄PF₆ (358 mg, 0.96 mmol) and 3,5-diphenyl-pyrazole (212 mg, 0.96 mmol) were dissolved in 5 mL acetone. The mixture was stirred to provide a clear, slightly green solution. The NEt₃ (0.16 mL 116 mg, 1.15 mmol) was added to stirred solution by dropwise and formed a white precipitate, which was stirred for 30 min, then filtered, washed with acetone and vacuum-dried. Finally, the product was dissolved in DCM-Et₂O solution, and colorless crystals of **Cu₃NC^(Pz)** were

obtained in a yield of 80% (218 mg). $^1$H NMR (600 MHz, CDCl$_3$) δ 7.69 (d, $J = 7.5$ Hz, 12H), 7.20 (t, $J = 7.3$ Hz, 8H), 7.06 (t, $J = 7.5$ Hz, 12H), 6.76 (s, 4H). $^{13}$C NMR (151 MHz, CDCl$_3$) δ 155.1, 132.6, 128.6, 128.0, 126.6, 102.2.

### General procedure for Cu$_3$NC$^{(NHC)}$-catalyzed the A$^3$ coupling reaction

Under N$_2$ atmosphere, the catalyst of Cu$_3$NC$^{(NHC)}$ (2.5 mol%), activated MS 4 Å sieves (600 mg) and dry DCM (1.5 mL) were added into the dry Schlenk tube. The reaction mixture was stirred at room temperature for 5 min. Aldehydes 1 (0.1 mmol or 0.2 mmol) and alkynes 2 (0.2 mmol or 0.1 mmol) were then added to the stirred mixture. The reaction mixture was stirred at room temperature for 5 minutes. Amines 3 (0.3 mmol, 1.5 eq) and dry DCM (0.5 mL, 0.1 M) were added into the stirred mixture of the Schlenk tube. The reactions were stirred at room temperature for 12 h. The reactions were monitored by TLC. When aldehydes 1 and alkynes 2 were consumed, the reactions were quenched and concentrated. The crude products were then purified by column chromatography to give the target molecules.

### General procedure for Cu$_3$NC$^{(NHC)}$-catalyzed the redox-A$^3$ coupling reaction

Under N$_2$ atmosphere, the catalyst of Cu$_3$NC$^{(NHC)}$ (0.25 mol%), activated MS 4 Å sieves (600 mg) and dry DCM (1.5 mL) were added into the dry Schlenk tube. The reaction mixture was stirred at room temperature for 5 min. Aldehydes 1 (0.2 mmol, 1.0 eq.) and alkynes 2 (0.2 mmol, 1.0 eq.) were then added to the stirred mixture. The reaction mixture was stirred at room temperature for 5 minutes. The 1,2,3,4-tetrahydroisoquinoline 3 (0.3 mmol, 1.5 eq.) and dry DCM (0.5 mL, 0.1 M) were added into the stirred mixture of the Schlenk tube. The reactions were stirred at room temperature for 40 h. The reactions were monitored by TLC. When aldehydes 1 and alkynes 2 were consumed, the reactions were quenched and concentrated. The crude products were then purified by column chromatography to give the target molecules.

### General procedure for Cu$_3$NC$^{(NHC)}$-catalyzed the A$^3$ coupling reaction in 2 mmol scale

Under N$_2$ atmosphere, the catalyst of Cu$_3$NC$^{(NHC)}$ (46 mg, 1.2 mol%), activated MS 4 Å sieves (5.0 g) and dry DCM (15.0 mL) were added into the dry Schlenk tube. The reaction mixture was stirred at room temperature for 5 min. Benzaldehyde 1 (213 mg, 2.0 mmol, 1.0 eq.) and phenylacetylene 2 (205.0 mg, 2.0 mmol, 1.0 eq. were then added to the stirred mixture. The reaction mixture was stirred at room temperature for 5 min. Dibenzylamine 3 (592.0 mg, 3.0 mmol, 1.5 eq.) and dry DCM (5.0 mL, 0.1 M) were added into the stirred mixture of the Schlenk tube. The reaction was stirred at room temperature for 12 h. The reactions were monitored by TLC. When Benzaldehyde 1 and Phenylacetylene 2 were consumed, the reaction was quenched and concentrated. The crude product was then purified by column chromatography (PE-EA, v/v 10/1) to give the N, N-dibenzyl-1,3-diphenylprop-2-yn-1-amine (4a) as colorless oil with overall isolated yield: 95% (735.7 mg).

### General procedure for Cu$_3$NC$^{(NHC)}$-catalyzed the redox-A$^3$ coupling reaction in 2 mmol scale

Under N$_2$ atmosphere, the catalyst of Cu$_3$NC$^{(NHC)}$ (9.2 mg, 0.25 mol%), activated MS 4 Å sieves (5.0 g) and dry DCM (15.0 mL) were added into the dry Schlenk tube. The reaction mixture was stirred at room temperature for 5 min. Benzaldehyde 1 (213 mg, 2.0 mmol, 1.0 eq.) and phenylacetylene 2 (205.0 mg, 2.0 mmol, 1.0 eq.) were then added to the stirred mixture. The reaction mixture was stirred at room temperature for 5 min. The 1,2,3,4-Tetrahydroisoquinoline 3 (400.0 mg, 3.0 mmol, 1.5 eq.) and dry DCM (5.0 mL, 0.1 M) were added into the stirred mixture of the Schlenk tube. The reaction was stirred at room temperature for 40 h. The reactions were monitored by TLC. When Benzaldehyde 1 and Phenylacetylene 2 were consumed, the reaction

was quenched and concentrated. The crude product was then purified by column chromatography (PE-EA, v/v 10/1) to give the 2-benzyl-1-(phenylethynyl)−1,2,3,4-tetrahydroisoquinoline (5a) as pale-yellow oil with overall isolated yield: 96% (620.5 mg).

### Recycling procedure for Cu$_3$NC$^{(NHC)}$-catalyzed the A$^3$ coupling reaction

Under N$_2$ atmosphere, the catalyst of Cu$_3$NC$^{(NHC)}$ (2.5 mol%), activated MS 4 Å sieves (600 mg) and dry DCM (1.5 mL) were added into the dry Schlenk tube. The reaction mixture was stirred at room temperature for 5 minutes. Aldehydes 1 (0.2 mmol, 1.0 eq.) and alkynes 2 (0.2 mmol, 1.0 eq.) were then added to the stirred mixture. The reaction mixture was stirred at room temperature for 5 minutes. Amines 3 (0.3 mmol, 1.5 eq.) and dry DCM (0.5 mL, 0.1 M) were added into the stirred mixture of the Schlenk tube. The reaction was stirred at room temperature for 12 h. Conversion and yield were determined by $^1$H NMR using 1,3,5-trimethoxybenzene as an internal standard. Then the reactants [aldehydes 1 (0.2 mmol, 1.0 eq.), alkynes 2 (0.2 mmol, 1.0 eq.), and amines 3 (0.3 mmol, 1.5 eq.)] were injected to the reaction system five consecutive times, subsequently. Every conversion and yield were determined by $^1$H NMR using 1,3,5-trimethoxybenzene as an internal standard.

### Recycling procedure for Cu$_3$NC$^{(NHC)}$-catalyzed the redox-A$^3$ coupling reaction

Under N$_2$ atmosphere, the catalyst of Cu$_3$NC$^{(NHC)}$ (0.25 mol%), activated MS 4 Å sieves (600 mg) and dry DCM (1.5 mL) were added into the dry Schlenk tube. The reaction mixture was stirred at room temperature for 5 min. Aldehydes 1 (0.2 mmol, 1.0 eq.) and alkynes 2 (0.2 mmol, 1.0 eq.) were then added to the stirred mixture. The reaction mixture was stirred at room temperature for 5 min. The 1,2,3,4-Tetrahydroisoquinoline 3 (0.3 mmol, 1.5 eq.) and dry DCM (0.5 mL, 0.1 M) were added into the stirred mixture of the Schlenk tube. The reaction was stirred at room temperature for 40 h. Conversion and yield were determined by $^1$H NMR using 1,3,5-trimethoxybenzene as an internal standard. Then the reactants [aldehydes 1 (0.2 mmol, 1.0 eq.), alkynes 2 (0.2 mmol, 1.0 eq.) and amines 3 (0.3 mmol, 1.5 eq.)] were injected to the reaction system two consecutive times, subsequently. Every conversion and yield were determined by $^1$H NMR using 1,3,5-trimethoxybenzene as an internal standard.

### Catalytic mechanism studies by density functional theory (DFT) calculations

DFT calculations were conducted using the Gaussian 16 package, specifically at the B3LYP-D3 level of theory, where dispersion interactions were considered[63–68]. For carbon, oxygen, hydrogen, and nitrogen atoms, the 6-31 G* basis set was employed, while the 6-311++G** basis set was utilized for copper[69,70]. To account for solvation effects, single-point calculations were performed using the solvation model based on density (SMD) in dichloromethane[71]. Vibrational frequency calculations were conducted to verify that a transition state exhibits only one imaginary frequency and a local minimum displays no imaginary frequency. Transition states connecting the relevant minima were further investigated through intrinsic reaction coordinate (IRC) calculations[72–74]. Corrections of −2.6 (or 2.6) kcal/mol were applied for transformations involving the conversion of two molecules to one (or one molecule to two) to obtain relative Gibbs energies at 298 K. This approach has been used in numerous previous studies to minimize the overestimation of entropy contributions[75–83].

According to the energetic span model, the schematic illustration of the energy profile given the A$^3$ coupling reaction and the redox-A$^3$ coupling reaction in Fig. 6, including the energy profile for an additional catalytic cycle (THIQs as the amine substrate). TS$_{6-7}$, TS$_{9-10}$, Int7, and Int9 represent rate-determining transition states and intermediates, respectively. We could also view the first Int7 to the second

TS$_{6-7}$ as a catalytic cycle from Supplementary Fig. 36. The first TS$_{9-10}$ to the second Int9 could be viewed as a catalytic cycle from Supplementary Fig. 37. Therefore, the overall barrier of A$^3$ coupling and redox-A$^3$ coupling could be calculated by $\Delta G_1 + \Delta G_2$ and $\Delta G_3 + \Delta G_4$, respectively. We obtained $\Delta G_1 = 27.6$ kcal mol$^{-1}$, $\Delta G_2 = 5.8$ kcal mol$^{-1}$, $\Delta G_3 = 19.8$ kcal mol$^{-1}$ and $\Delta G_4 = 6.7$ kcal mol$^{-1}$ from Fig. 6.

## Data availability

All other data are available from the corresponding author upon request. All data needed to evaluate the conclusions in the paper are present in the paper and/or the Supplementary Materials (including Supplementary Figs. 1–226, details of the chemicals, instrumentation, synthesis, characterization, DFT and X-ray crystal details for Cu$_3$NC$^{(NHC)}$ (CIF) and Cu$_3$NC$^{(Pz)}$ (CIF)). The source data for Figs. 5 and 6 are provided with this paper. The X-ray crystallographic coordinates for the structures reported in this article have been deposited at the Cambridge Crystallographic Data Centre (CCDC), under deposition number CCDC 2268919 (Cu$_3$NC$^{(NHC)}$) 2268920 (Cu$_3$NC$^{(Pz)}$). These data can be obtained free of charge via www.ccdc.cam.ac.uk/data_request/cif. Source data are provided with this paper.

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

## Acknowledgements

This work was supported by the National Natural Science Foundation of China (No. 21825106, S.-Q.Z.; 92061201, S.-Q.Z.), the China Postdoctoral Science Foundation (2022M722864, T.J.), Zhongyuan Thousand Talents (Zhongyuan Scholars) Program of Henan Province (234000510007, S.-Q.Z.), the Excellent Youth Foundation of Henan Scientific Committee (232300421022, X.-Y.D.) and Zhengzhou University.

## Author contributions

S.-Q.Z. and T.J. conceived and designed the idea. T.J. prepared nanocluster catalysts and conducted the organic transformation experiments. Y.-X.L., X.-H.M., and J.A. assisted with the experiments and characterizations. T.J. and X.-Y.D. wrote the manuscript. M.-M.Z. helped with the manuscript revising and data analysis. S.-Q.Z., T.J., and X.-Y.D. discussed the results and prepared the manuscript. All the authors reviewed and contributed to this paper.

## Competing interests

The authors declare no competing interests.
