## [Peer Review File · Nature Communications]

Atomically Precise Ultrasmall Copper Cluster for Room-Temperature Highly Regioselective Dehydrogenative CouplingReviewers' Comments:

Reviewer #1:

Remarks to the Author:

In this manuscript, Zang and co-workers designed a new tridentate N-heterocyclic carbene ligand and developed a novel rigid-flexible-coupled copper cluster $[\text{Cu}_3(\text{NHC})_3(\text{PF}_6)_3]$ ($\text{Cu}_3\text{NC}(\text{NHC})$) on gram scale. The ultrasmall clusters were highly stable and have been successfully used to catalyze three-component dehydrogenative coupling reactions under mild conditions. The strength of this system was demonstrated by its high efficiency, broad substrate scopes, and excellent regioselectivity. Mechanistic studies illustrated that the regioselectivity of the process is associated with the energy of the ion pair with copper acetylide and iminium cation. The reactions were nicely developed and the supporting information is well-organized. In my opinion, this manuscript could be published in Nature Communications after the following minor revisions have been addressed.

- 1) In the A3 coupling reaction, how about the reaction with primary amines ?
- 2) In the redox-A3 coupling reaction, only 1,2,3,4-tetrahydroisoquinoline (THIQs) was used as the coupling partner. How about the reaction with other amines (such as 3be, 3bm, 3bn, 3bp) under the conditions in Figure 4.
- 3) The NMR spectrum of compounds (4k, 4u, 4aj, 4ba, 4be, 4bh, 4bp...) should be collected with samples in higher concentration.
- 4) The author should check the manuscript carefully. For example, $\text{Cu}_3\text{NC}(\text{NCH})$ should be replaced by $\text{Cu}_3\text{NC}(\text{NHC})$ (Figures 2d and 2e).
- 5) The following paper should be added to the corresponding parts:
[10.3390/molecules24071216](https://doi.org/10.3390/molecules24071216)

Reviewer #2:

Remarks to the Author:

In this paper, Zang and coauthors report synthesis of an atomically precise and rigid-flexible-coupled $\text{Cu}_3\text{NC}(\text{NHC})$ nanocluster in the gram scale, supported by the designed tridentate N-heterocyclic carbene ligand. After systematically characterizing the $\text{Cu}_3\text{NC}(\text{NHC})$ catalyst, the $\text{Cu}_3\text{NC}(\text{NHC})$ is applied to achieve highly all-in-one dehydrogenative coupling functionalization under mild conditions, including excellent regioselectivity (up to 100%), high efficiency (TON = 3800 and 1840), high yield (up to 99%) and broad substrate scopes (85 examples), etc. More importantly, they found that the shell ligand with the DLL endows $\text{Cu}_3\text{NC}(\text{NHC})$ with rigidity and flexibility, which could enhance its stability, present a dynamically catalytic centre and enhance the regioselectivity of dehydrogenative coupling transformations through weak interactions. For the first time, atomically precise copper cluster showing outstanding catalytic performance and regioselectivity for all-in-one dehydrogenative coupling at room temperature could be achieved in metal cluster systems, which provided a novel efficient strategy to construct metal cluster-based catalysts for future catalytic functional applications. This work combines the excellent catalytic performance characteristics, dynamic catalytic centres, DLLs, ultrahigh stability and atomically precise structures in a single copper cluster protected by N-heterocyclic carbene ligands. The precise structure-activity relationship of $\text{Cu}_3\text{NC}(\text{NHC})$ catalyst at the atomic level is also clarified by Mechanistic studies and DFT. In addition, all conclusions are supported by the experimental data presented. In a word, this work has high originality and represents a milestone on the road to metal nanocluster-catalyzed organic reactions. I would like to recommend its publication in the respected journal, Nature Communications, after some minor revisions.

1. The as-fabricated copper nanocluster with tridentate N-heterocyclic carbene ligands showed excellent catalytic performance. I suggest the catalytic performance characteristic of the shell ligand could be listed in the table.
2. The dehydrogenative coupling transformations are catalyzed by the copper clusters to work well under basic condition. But the text only shows that stability of Cu₃NC(BINAP) and Cu₃NC(Pz) can retain intact in organic solvents. How stable are these clusters under basic condition?
3. The authors have evaluated the stability of Cu₃NC(NHC) and Cu₃NC(BINAP) in DCM by using time-dependent UV-vis spectra. However, only time-dependent ¹H NMR spectra were used to demonstrate the stability of Cu₃NC(Pz) (the detection limit of UV-vis spectra which is more sensitive than the detection limit of NMR). The Cu₃NC(Pz) catalyst should be further characterized by time-dependent UV-vis spectra measurements.
4. Please correct the error: the Table 2 "[Cu] catal. (0.005 mmol, 0.25 mol%)" mistake on "0.005 mmol", which is different from the Fig. 4 "Cu₃NC(NHC) (0.0005 mmol, 0.25 mol%)". Please check and confirm it.
5. Please report the ¹³C NMR data as one digit after the decimal point: "CD₃CN (δ 118.26 ppm) for ¹³C NMR" in the Supplementary Information.
6. Please correct the error: the figure notes of Supplementary Figure 31 ("A₃-Product and redox-A₃ Product") in the Supplementary Information, S24.

Reviewer #3:

Remarks to the Author:

In this manuscript, Zeng and co-workers designed a novel tridentate N-heterocyclic carbene ligand and synthesized a Cu₃ cluster [Cu₃(NHC)₃(PF₆)₃] (Cu₃NC(NHC)) for room-temperature dehydrogenative coupling. The molecular-level structure and the structure-activity relationship of Cu₃ nanoclusters are clarified, which provides a reference for the application of metal nanoclusters in homogeneous organic transformation, and can promote more exploration of organic reactions catalyzed by metal nanoclusters. I believe this work is valuable for researchers from fields of organic chemistry, organometallic synthesis and nanocatalysis, which can be published in Nature Commun. after addressing the following issues.

1. The authors state that activated MS 4A sieves alone could not catalyze the reaction, but the lack of MS 4A sieves would decrease the yield of product. Have the Cu₃ nanoclusters been captured within the molecular sieves or been adsorbed on the surface of MS during catalytic reaction? What is the role of MS 4A sieves in this catalytic system?
2. The authors' catalytic cycle experiments do not seem reasonable. From the author's description of the catalytic cycle experiment, it can be seen that the recyclability of the catalyst is evaluated by adding new reactants to the solution containing Cu₃ catalysts for several times without the separation of catalyst from reaction solution. For Cu₃NC(NHC) catalyzed A₃ coupling reaction, this experimental process is relatively reasonable, since the yield is close to 100% after each cycle. For Cu₃NC(NHC) catalyzed Redox-A₃ coupling reaction, each round of reaction yield is about 55%. There were still unreacted reactants in the system after the reaction, hence it is not appropriate to follow the catalytic cycle assessment procedure described in the manuscript.
3. For catalytic recycling assessment, it is usually necessary to separate the catalyst from the catalytic system. It is suggested that the author try to load Cu₃ clusters on activated carbon and assess the recyclability of Cu catalysts by separating the reaction products and catalysts after each reaction round.
4. The Cu₃NC(NHC) proposed in the abstract and introduction has the dual feature of rigidity and flexibility. In the following introduction, the attribute of rigidity is clearly introduced, but the length of

flexibility is not clear. It is suggested that the author modify it briefly.

5. It is suggested that the authors avoid the use of redundant function words, such as the repeated use of "Furthermore" in page 9 paragraph 1. There are also some format flaws in the article, the author should review the article in detail again.

Responses to Reviewers

Reviewer #1: (minor revisions)

Remarks: In this manuscript, Zang and co-workers designed a new tridentate N-heterocyclic carbene ligand and developed a novel rigid-flexible-coupled copper cluster $[\text{Cu}_3(\text{NHC})_3(\text{PF}_6)_3]$ ($\text{Cu}_3\text{NC}^{\text{(NHC)}}$) on gram scale. *The ultrasmall clusters were highly stable and have been successfully used to catalyze three-component dehydrogenative coupling reactions under mild conditions. The strength of this system was demonstrated by its high efficiency, broad substrate scopes, and excellent regioselectivity. Mechanistic studies illustrated that the regioselectivity of the process is associated with the energy of the ion pair with copper acetylide and iminium cation. The reactions were nicely developed and the supporting information is well-organized. In my opinion, this manuscript could be published in Nature Communications after the following minor revisions have been addressed.*

Response: We thank **Reviewer 1** for the positive comments very much and have addressed the following issues.

Comment 1: In the A^3 coupling reaction, how about the reaction with primary amines?

Response: Thanks for the reviewer's comment. As suggested, we have tested the primary amines, aniline, p-toluidine, 4-fluoroaniline, amantadine, thiophene-2-ethylamine, and 1-octylamine. The primary amines are considered to be inert substrates for A^3 -coupling transformations under mild conditions (reference, *Chem. Eur. J.* **16**, 3281–3284 (2010); *Angew. Chem. Int. Ed.* **48**, 3116–3120, (2009)), no reaction was observed under optimal conditions. We have added the corresponding discussion in the revised manuscript for clarification.

Comment 2: In the redox- A^3 coupling reaction, only 1,2,3,4-tetrahydroisoquinoline (THIQs) was used as the coupling partner. How about the reaction with other amines (such as 3be, 3bm, 3bn, 3bp) under the conditions in Figure 4.

Response: This is a great point. Thank the reviewer so much for the great suggestion! We have tested N-phenylbenzylamine (3be), 1-ethylpiperazine (3bm), 1-phenylpiperazine (3bn), N-methylbenzylamine (3bp), and pyrrolidine (3bc) as the amine substrates in the redox- A^3 coupling reaction. The redox- A^3 coupling products were not observed, indicating that *in situ-generated exo-iminium ions* are not always easy to isomerize to *endo-iminium ions*. We have included this result in the revised manuscript.

Comment 3: The NMR spectrum of compounds (4k, 4u, 4aj, 4ba, 4be, 4bh, 4bp) should be collected with samples in higher concentration.

Response: Thanks very much for the reviewer's comment. We have made all the revisions of the NMR spectrum of compounds (4k, 4u, 4aj, 4ba, 4be, 4bh, 4bp) as suggested.

Supplementary Figure 69. ¹H NMR spectrum of compound 4k.

Supplementary Figure 70. ¹³C NMR spectrum of compound 4k.

Supplementary Figure 89. ¹H NMR spectrum of compound 4u.

Supplementary Figure 90. ¹³C NMR spectrum of compound 4u.

Supplementary Figure 121. ¹H NMR spectrum of compound 4aj.

Supplementary Figure 122. ¹³C NMR spectrum of compound 4aj.

Supplementary Figure 156. ¹H NMR spectrum of compound 4ba.

Supplementary Figure 157. ¹³C NMR spectrum of compound 4ba.

Supplementary Figure 164. $^1\text{H NMR}$ spectrum of compound **4be**.

Supplementary Figure 165. $^{13}\text{C NMR}$ spectrum of compound **4be**.

Supplementary Figure 170. ¹H NMR spectrum of compound 4bh.

Supplementary Figure 171. ¹³C NMR spectrum of compound 4bh.

Supplementary Figure 188. ¹H NMR spectrum of compound 4bp.

Supplementary Figure 189. ¹³C NMR spectrum of compound 4bp.

Comment 4: The author should check the manuscript carefully. For example, $\text{Cu}_3\text{NC}^{(\text{NCH})}$ should be replaced by $\text{Cu}_3\text{NC}^{(\text{NHC})}$ (Figures 2d and 2e).

Response: Thanks for the correction. We have corrected the corresponding mistake in the revised manuscript.

Comment 5: The following paper should be added to the corresponding parts: 10.3390/molecules24071216.

Response: Thanks a lot for the reviewer's suggestion. As suggested, we have cited the paper in the revised manuscript.

Reviewer #2: (minor revisions)

Remarks: In this paper, Zang and coauthors report synthesis of an atomically precise and rigid-flexible-coupled $\text{Cu}_3\text{NC}^{(\text{NHC})}$ nanocluster in the gram scale, supported by the designed tridentate N-heterocyclic carbene ligand. After systematically characterizing the $\text{Cu}_3\text{NC}^{(\text{NHC})}$ catalyst, the $\text{Cu}_3\text{NC}^{(\text{NHC})}$ is applied to achieve highly all-in-one dehydrogenative coupling functionalization under mild conditions, including excellent regioselectivity (up to 100%), high efficiency (TON = 3800 and 1840), high yield (up to 99%) and broad substrate scopes (85 examples), etc. *More importantly, they found that the shell ligand with the DLL endows $\text{Cu}_3\text{NC}^{(\text{NHC})}$ with rigidity and flexibility, which could enhance its stability, present a dynamically catalytic centre and enhance the regioselectivity of dehydrogenative coupling transformations through weak interactions. For the first time, atomically precise copper cluster showing outstanding catalytic performance and regioselectivity for all-in-one dehydrogenative coupling at room temperature could be achieved in metal cluster systems, which provided a novel efficient strategy to construct metal cluster-based catalysts for future catalytic functional applications.* This work combines the excellent catalytic performance characteristics, dynamic catalytic centres, DLLs, ultrahigh stability and atomically precise structures in a single copper cluster protected by N-heterocyclic carbene ligands. The precise structure–activity relationship of $\text{Cu}_3\text{NC}^{(\text{NHC})}$ catalyst at the atomic level is also clarified by Mechanistic studies and DFT. In addition, all conclusions are supported by the experimental data presented. *In a word, this work has high originality and represents a milestone on the road to metal nanocluster-catalyzed organic reactions. I would like to recommend its publication in the respected journal, Nature Communications, after some minor revisions.*

Response: We thank **Reviewer 2** for the positive comments very much and have addressed the following issues.

Comment 1: The as-fabricated copper nanocluster with tridentate N-heterocyclic carbene ligands showed excellent catalytic performance. I suggest the catalytic performance characteristic of the shell ligand could be listed in the table.

Response: This is a good point. Thank the reviewer for this great suggestion! We have tested the shell ligand as a catalyst to catalyze the three-component dehydrogenative coupling reactions under optimal conditions. The A^3 coupling products and the redox- A^3 coupling products were not obtained, indicating that the shell ligand has no catalytic performances in the three-component dehydrogenative coupling transformations. We have included these results in the revised manuscript.

Comment 2: The dehydrogenative coupling transformations are catalyzed by the copper clusters

to work well under basic condition. But the text only shows that stability of $\text{Cu}_3\text{NC}^{(\text{BINAP})}$ and $\text{Cu}_3\text{NC}^{(\text{Pz})}$ can retain intact in organic solvents. How stable are these clusters under basic condition?

Response: Thanks very much for the reviewer's suggestions. The time-dependent UV-vis spectrum of $\text{Cu}_3\text{NC}^{(\text{BINAP})}$ (Supplementary Figure 19) and the time-dependent ^1H NMR spectrum of $\text{Cu}_3\text{NC}^{(\text{Pz})}$ (Supplementary Figure 25) were tested as suggested and showed no discernible change in basic solvent, suggesting that $\text{Cu}_3\text{NC}^{(\text{BINAP})}$ and $\text{Cu}_3\text{NC}^{(\text{Pz})}$ could retain intact under basic conditions. These results have been included in the revised manuscript.

Supplementary Figure 19. UV-vis spectra tracking of $\text{Cu}_3\text{NC}^{(\text{BINAP})}$ - HNBN_2 (1:3) in DCM (0-72 h).

Supplementary Figure 25. ^1H NMR spectra tracking of $\text{Cu}_3\text{NC}^{(\text{Pz})}$ - HNBN_2 (1:3) in CD_3Cl (0-72 h).

Comment 3: The authors have evaluated the stability of $\text{Cu}_3\text{NC}^{\text{(NHC)}}$ and $\text{Cu}_3\text{NC}^{\text{(BINAP)}}$ in DCM by using time-dependent UV-vis spectra. However, only time-dependent ^1H NMR spectra were used to demonstrate the stability of $\text{Cu}_3\text{NC}^{\text{(Pz)}}$ (the detection limit of UV-vis spectra which is more sensitive than the detection limit of NMR). The $\text{Cu}_3\text{NC}^{\text{(Pz)}}$ catalyst should be further characterized by time-dependent UV-vis spectra measurements.

Response: Thanks for the reviewer's comment. As suggested, the time-dependent UV-vis spectra of $\text{Cu}_3\text{NC}^{\text{(Pz)}}$ were tested, showing that the UV-vis absorption behavior of $\text{Cu}_3\text{NC}^{\text{(Pz)}}$ in DCM was unchanged after 72 h of treatment, which further declares the excellent stability of $\text{Cu}_3\text{NC}^{\text{(Pz)}}$ in organic solvent (Supplementary Figure 20). We have included these results in the revised manuscript.

Supplementary Figure 20. UV-vis spectra tracking of $\text{Cu}_3\text{NC}^{\text{(Pz)}}$ in DCM (0-72 h).

Comment 4: Please correct the error: the Table 2 “[Cu] catal. (0.005 mmol, 0.25 mol%)” mistake on “0.005 mmol”, which is different from the Fig. 4 “ $\text{Cu}_3\text{NC}^{\text{(NHC)}}$ (0.0005 mmol, 0.25 mol%)”. Please check and confirm it.

Response: Thanks for your reminder. We have corrected the corresponding mistake in the revised manuscript.

Comment 5: Please report the ^{13}C NMR data as one digit after the decimal point: “ CD_3CN (δ 118.26 ppm) for ^{13}C NMR” in the Supplementary Information.

Response: Thanks for the comments. We have made the revision as suggested. Thank you so much for helping with the manuscript.

Comment 6: Please correct the error: the figure notes of Supplementary Figure 31 (“ A_3 -Product and redox- A_3 Product”) in the Supplementary Information, S24.

Response: Thanks a lot for the comments. We have corrected the corresponding mistake in the revised manuscript.

Reviewer #3: (minor revisions)

Remarks: In this manuscript, Zang and co-workers designed a novel tridentate N-heterocyclic carbene ligand and synthesized a Cu₃ cluster [Cu₃(NHC)₃(PF₆)₃] (Cu₃NC^(NHC)) for room-temperature dehydrogenative coupling. The molecular-level structure and the structure-activity relationship of Cu₃ nanoclusters are clarified, which provides a reference for the application of metal nanoclusters in homogeneous organic transformation, and can promote more exploration of organic reactions catalyzed by metal nanoclusters. I believe this work is valuable for researchers from fields of organic chemistry, organometallic synthesis and nanocatalysis, which can be published in Nature Communications after addressing the following issues.

Response: We thank **Reviewer 3** for the positive comments very much and have addressed the following issues.

Comment 1: The authors state that activated MS 4Å sieves alone could not catalyze the reaction, but the lack of MS 4Å sieves would decrease the yield of product. Have the Cu₃ nanoclusters been captured within the molecular sieves or been adsorbed on the surface of MS during catalytic reaction? What is the role of MS 4Å sieves in this catalytic system?

Response: This comment is an excellent point! Thank the reviewer for this insightful comment. We have conducted the additional control experiments following this suggestion. Under N₂ atmosphere, the Cu₃NC^(NHC) (0.005 mmol) was dissolved in 2 mL dried DCM, which was added dropwise into 5 mL dried DCM solution of 600 mg MS 4Å sieves under vigorous stirring. After stirring for 6 h at room temperature, the MS 4Å sieves were washed with dried DCM three times and then dried in vacuum (references: *Angew. Chem. Int. Ed.* **58**, 17731–17735 (2019); *Angew. Chem. Int. Ed.* **60**, 970–975 (2021)). The specially treated MS 4Å sieves, as a kind of adsorption drying agent and catalyst, were tested to catalyze A³ coupling reaction, affording 14% yields for 5 h and 31% yields for 12 h. Compared to the catalytic performance of the specially treated MS 4Å sieves, the activated MS 4Å sieves alone could not catalyze the transformation. These results indicate that a little Cu₃NC^(NHC) could be adsorbed on the MS 4Å sieves to provide poor yields of the A³ coupling product.

In this work, in the three-component dehydrogenative coupling reaction systems, the activated MS 4Å sieves, as a kind of adsorption drying agent, could remove by-product water in time, enhancing dehydrogenative coupling reactions rate and improving the yields of propargylamines and C1-propargylamines beyond 95%. We have included this result in the revised manuscript.

Comment 2: The authors' catalytic cycle experiments do not seem reasonable. From the author's description of the catalytic cycle experiment, it can be seen that the recyclability of the catalyst is evaluated by adding new reactants to the solution containing Cu₃ catalysts for several times without the separation of catalyst from reaction solution. For Cu₃NC^(NHC) catalyzed A³ coupling reaction, this experimental process is relatively reasonable, since the yield is close to 100% after each cycle. For Cu₃NC^(NHC) catalyzed Redox-A³ coupling reaction, each round of reaction yield is about 55%. There were still unreacted reactants in the system after the reaction, hence it is not appropriate to follow the catalytic cycle assessment procedure described in the manuscript.

Response: Thank the reviewer very much for the comment. As **Reviewer 3** suggested, the catalytic cycle experiments of redox-A³ coupling reaction were re-investigated. The reaction time was extended from 12 h to 40 h in the catalytic cycle experiments. Under the optimized conditions

of redox-A³ coupling transformation, the investigation found that the Cu₃NC^(NHC) catalyst displayed excellent stability and reusability in redox-A³ coupling reaction. The redox-A³ coupling reaction was conducted three times with the Cu₃NC^(NHC) catalyst. Nearly quantitative yields were obtained in all transformations. We have revised the manuscript as suggested.

Supplementary Figure 43. Recyclability of the Cu₃NC^(NHC) catalyzed Redox-A³ coupling reaction in term of yield (40 h).

Comment 3: For catalytic recycling assessment, it is usually necessary to separate the catalyst from the catalytic system. It is suggested that the author try to load Cu₃ clusters on activated carbon and assess the recyclability of Cu catalysts by separating the reaction products and catalysts after each reaction round.

Response: This is a very good point. Thanks very much for the reviewer's suggestion! As suggested, the control experiments were tested. Under N₂ atmosphere, the Cu₃NC^(NHC) (0.005mmol) was dissolved in 2 mL dried DCM, which was added dropwise into 5 mL dried DCM solution of 100 mg activated carbon (Vulcan XC-72 carbon black) under vigorous stirring. After stirring for 6 h at room temperature, the solution was centrifuged at 8000 rpm for 5 minutes, and the activated carbon was washed with dried DCM three times and then dried in vacuum (reference: *Angew. Chem. Int. Ed.* **58**,17731–17735 (2019); *Angew. Chem. Int. Ed.* **60**, 970–975 (2021)).

The Cu₃NC^(NHC)/XC-72 as catalyst was tested to catalyze the A³ coupling reaction under optimized conditions, affording the desired typical propargylamine product with excellent yields (98%) for the first time. The activated MS 4Å sieves were taken out and the reaction mixture was centrifuged at 8000 rpm for 5 minutes and purified to give the target molecule. The Cu₃NC^(NHC)/XC-72 was washed with hexane three times then dried in vacuum and used again in the next round of recycling. The result indicates that the Cu₃NC^(NHC)/XC-72 catalyzed the A³ coupling reaction under the optimized conditions, affording propargylamine products with poor yields (35%) for the second time. To investigate the origin of the decrease in A³ coupling reaction yield, the reaction process was monitored by the *in situ* ¹H NMR spectrum that the Cu₃NC^(NHC)/XC-72 as a catalyst, for the first time, catalyzed the A³ coupling reaction under the

optimized conditions. The characteristic peaks of $\text{Cu}_3\text{NC}^{\text{(NHC)}}$ were found in the *in situ* ^1H NMR spectrum (Supplementary Figure 30), indicating that $\text{Cu}_3\text{NC}^{\text{(NHC)}}$ is very easy to separate from activated carbon to lead to the decrease in A^3 coupling reaction yield on the second time. The corresponding description has been added in the revised manuscript.

Supplementary Figure 30. *In situ* ^1H NMR spectra tracking of the $\text{Cu}_3\text{NC}^{\text{(NHC)}}$ /XC-72 catalyzed A^3 coupling reaction for 12 h in DCM, for the first time.

Comment 4: The $\text{Cu}_3\text{NC}^{\text{(NHC)}}$ proposed in the abstract and introduction has the dual feature of rigidity and flexibility. In the following introduction, the attribute of rigidity is clearly introduced, but the length of flexibility is not clear. It is suggested that the author modify it briefly.

Response: As suggested by Reviewer 3, we added elucidation of the flexibility of $\text{Cu}_3\text{NC}^{\text{(NHC)}}$ cluster in the discussion.

The flexibility of $\text{Cu}_3\text{NC}^{\text{(NHC)}}$ originates in the flexible bonding and dissociation between pyridine groups of ligands and Cu centers. When the original pyridine moieties in DLL ligands were dissociated from Cu_3 core in the catalytic systems, pyridines could rotate freely to decrease the steric hindrance around the copper center, making the catalytic center exposed to the catalytic substrate. While the reaction was completed, pyridine groups reversibly coordinated to Cu atoms, restoring the original structure of the cluster. The flexibility ensures the stability and reversibility of this copper cluster catalyst.

Comment 5: It is suggested that the authors avoid the use of redundant function words, such as the repeated use of "Furthermore" in page 9 paragraph 1. There are also some format flaws in the article, the author should review the article in detail again.

Response: Following this suggestion, we have double-checked the manuscript and corrected the format flaws.

Reviewers' Comments:

Reviewer #1:

Remarks to the Author:

Overall, the authors have done a good job and considered seriously all comments made. The new results have addressed my points and I support this paper for publication in Nature Communications.

Reviewer #2:

Remarks to the Author:

The authors have addressed all my concerns, and I am very enthusiastic to recommend the acceptance of this paper.

Reviewer #3:

Remarks to the Author:

The revised version can be accepted.

Responses to Editor

Remarks: *We therefore invite you to revise your paper one last time to address the remaining concerns of our reviewers and our editorial requests in the attached documents. At the same time we ask that you edit your manuscript to comply with our policies and formatting requirements and to maximise the accessibility and therefore the impact of your work.*

Response: We thank **Editor** for the positive comments very much and we have revised the manuscript as requested.

Responses to Reviewers

Reviewer #1:

Remarks: *Overall, the authors have done a good job and considered seriously all comments made. The new results have addressed my points and I support this paper for publication in Nature Communications.*

Response: We thank **Reviewer 1** for the positive comments very much.

Reviewer #2:

Remarks: *The authors have addressed all my concerns, and I am very enthusiastic to recommend the acceptance of this paper.*

Response: We thank **Reviewer 2** for the positive comments very much.

Reviewer #3:

Remarks: *The revised version can be accepted.*

Response: We thank **Reviewer 3** for the positive comments very much.